# Memory-dictated dynamics of single-atom Pt on $CeO_2$ for CO oxidation

Zihao Zhang[1,2,6], Jinshu Tian[1,6], Yubing Lu[1], Shize Yang[3], Dong Jiang[2], Weixin Huang[2], Yixiao Li[2], Jiyun Hong[4], Adam S. Hoffman[4], Simon R. Bare[4], Mark H. Engelhard[1], Abhaya K. Datye[5] & Yong Wang[1,2] ✉

Single atoms of platinum group metals on $CeO_2$ represent a potential approach to lower precious metal requirements for automobile exhaust treatment catalysts. Here we show the dynamic evolution of two types of single-atom Pt ($Pt_1$) on $CeO_2$, i.e., adsorbed $Pt_1$ in $Pt/CeO_2$ and square planar $Pt_1$ in $Pt_{AT}CeO_2$, fabricated at 500 °C and by atom-trapping method at 800 °C, respectively. Adsorbed $Pt_1$ in $Pt/CeO_2$ is mobile with the in situ formation of few-atom Pt clusters during CO oxidation, contributing to high reactivity with near-zero reaction order in CO. In contrast, square planar $Pt_1$ in $Pt_{AT}CeO_2$ is strongly anchored to the support during CO oxidation leading to relatively low reactivity with a positive reaction order in CO. Reduction of both $Pt/CeO_2$ and $Pt_{AT}CeO_2$ in CO transforms $Pt_1$ to Pt nanoparticles. However, both catalysts retain the memory of their initial $Pt_1$ state after reoxidative treatments, which illustrates the importance of the initial single-atom structure in practical applications.

Single-atom catalysts (SACs) have been attracting widespread attention in the catalysis community for the past decade[1,2]. Among them, $CeO_2$-supported SACs are particularly interesting because of the oxygen storage capacity of $CeO_2$ and the ability of $CeO_2$ to intrinsically trap platinum group metals (PGMs: Pt, Pd, Rh, etc.) under high-temperature oxidative condition[3–6]. $CeO_2$-supported PGMs prepared by atom-trapping (AT) method at 800 °C have recently been reported to be promising sintering-resistant catalysts for the removal of vehicle criteria pollutants (e.g., CO, $NO_x$, and hydrocarbons)[7–9]. While the maximum atomic utilization can be realized for isolated PGMs (e.g., $Pt_1$), the intrinsic activity of $Pt_1$ is usually lower than Pt aggregates[10–12]. To circumvent this issue, $Pt_1$ on $CeO_2$ was transformed to more active Pt nanoparticles (NPs, <2 nm) via the treatment in reducing atmospheres (CO, $H_2$, or HCs) at elevated temperatures[10,13,14]. However, these agglomerated Pt NPs redisperse into less-active $Pt_1$ under an additional treatment in $O_2$ or even in a lean reaction condition at temperatures higher than 400 °C[15,16], which complicates their applications in practical exhaust gas treatment.

It has been reported that single-atom Pt, Pd, or Cu on $CeO_2$ fabricated by different annealing temperatures show various catalytic performances[13,17,18]. However, the origin of significant reactivity difference induced by different annealing temperatures is still unknown[4,7–10]. Although the dynamics of $Pt_1$ under reductive and oxidative conditions are both studied, the dynamic evolution of different types of $Pt_1$ under a real reaction condition is still missing. Understanding the initial $Pt_1$ structure and its dynamics under reaction conditions is of great importance to design more efficient $Pt_1$ or its derived active site for practical exhaust gas treatment. Therefore, two types of $Pt_1$ on $CeO_2$ catalysts were fabricated, one via treatment at 500 °C ($Pt/CeO_2$) and the second by atom-trapping method at 800 °C ($Pt_{AT}CeO_2$). The local structure and dynamic behavior of the two $Pt_1$ structures under CO oxidation condition were studied by in situ X-ray absorption spectroscopy (XAS), in situ infrared

[1]Institute for Integrated Catalysis, Pacific Northwest National Laboratory, Richland, WA 99354, USA. [2]The Gene and Linda Voiland School of Chemical Engineering and Bioengineering, Washington State University, Pullman, WA 99164, USA. [3]Eyring Materials Center, Arizona State University, Tempe, AZ 85257, USA. [4]Stanford Synchrotron Radiation Light Source, SLAC National Accelerator Laboratory, Menlo Park, CA 94025, USA. [5]Department of Chemical and Biological Engineering and Center for Micro-Engineered Materials, University of New Mexico, Albuquerque, NM 87131, USA. [6]These authors contributed equally: Zihao Zhang, Jinshu Tian. ✉e-mail: yong.wang@pnnl.gov

spectroscopy, quasi in situ X-ray photoelectron spectroscopy (XPS), and density functional theory (DFT) calculations. Both types of $Pt_1$ structures were studied under treatment in CO at 275 °C which led to the formation of Pt NPs in both $Pt/CeO_2$-CO and $Pt_{AT}CeO_2$-CO, followed by a reoxidative treatment at 500 °C to disintegrate the as-formed Pt NPs to form $Pt_1$ again in $Pt/CeO_2$-CO-$O_2$ and $Pt_{AT}CeO_2$-CO-$O_2$ (Fig. 1). The dynamics of $Pt_1$ under CO oxidation, reductive and oxidative treatments are investigated by comparing their CO oxidation activity, reaction kinetics, characterization results, and theoretical calculations.

## Results and discussion
### Single-atom $Pt_1$ structure in fresh $Pt/CeO_2$ and $Pt_{AT}CeO_2$
$Pt/CeO_2$ and $Pt_{AT}CeO_2$ with ~1 wt% Pt loading (Supplementary Table S1) were synthesized by two post-calcination temperatures of 500 and 800 °C in air, in which the calcination temperature of 800 °C represents a previously reported atom-trapping method[8]. Aberration-corrected high-angle annular dark-field scanning transmission electron microscopy (HAADF-STEM) images in Fig. 2a–d and Supplementary Fig. S1, and line-scanning results in the inset of Fig. 2d display that isolated $Pt_1$ atoms are present in $Pt/CeO_2$ and $Pt_{AT}CeO_2$. The powder X-ray diffraction (XRD) patterns of $Pt/CeO_2$ and $Pt_{AT}CeO_2$ in Supplementary Fig. S2 show only the diffraction peaks for fluorite $CeO_2$. Pt $L_3$-edge X-ray absorption near edge structure (XANES) spectroscopy in Fig. 2e, Supplementary Fig. S3 exhibits a white line intensity slightly lower than the $PtO_2$ ($Pt^{4+}$) reference, indicating a cationic $Pt^{\sigma+}$ nature ($\sigma < 4$)[19]. In contrast, Pt $4f$ X-ray photoelectron spectroscopy (XPS) in Fig. 2g displays a similar characteristic of $Pt^{2+}$ for two fresh samples[3]. The observed different Pt valences ($Pt^{2+}$ in XPS, near $Pt^{4+}$ in XANES) are mainly ascribed to various oxygen partial pressures in XANES (ambient air) and XPS (vacuum) measurement conditions[20]. Both the percentage of surface $Ce^{3+}$ and the defect-related O in $Pt/CeO_2$ and $Pt_{AT}CeO_2$ are similar (Fig. 2h, Supplementary Fig. S4). The extended X-ray absorption fine structure (EXAFS) results in Fig. 2f, Supplementary Fig. S3, display that the two samples are dominated by the first-shell Pt-O contribution, and the corresponding coordination number (CN) is $5.0 \pm 0.43$ for $Pt/CeO_2$, and $4.9 \pm 0.52$ for $Pt_{AT}CeO_2$ (Supplementary Table S2, Supplementary Figs. S5 and S6). The above ex situ characterizations suggest that the two fresh catalysts have the same atomically dispersed nature, similar Pt valence, similar Pt-O local coordination, and similar $Ce^{3+}$ and defect-related O percentage. However, their difference can be revealed by diffuse-reflectance infrared Fourier transform spectra with CO as a probe molecule (CO-DRIFTS) (Fig. 2i, Supplementary Fig. S7), which reveals a single IR band at ~2094 for $Pt/CeO_2$ and ~2089 $cm^{-1}$ for $Pt_{AT}CeO_2$ at 100 °C under CO oxidation condition, ascribed to CO linearly adsorbed on ionic Pt[21]. The different

vibration frequencies of adsorbed CO molecules can be tentatively assigned to their different CO-$Pt_1$ interactions under CO oxidation condition[22]. This implies the possible structural change of $Pt_1$ from ambient air to CO oxidation condition for $Pt/CeO_2$ or $Pt_{AT}CeO_2$. Previous studies reported that $Pt_1$ on $CeO_2$ synthesized by atom-trapping method holds a square planar structure[7,21,23]; however, $Pt_1$ structure synthesized at low calcination temperature is less discussed.

### CO oxidation activity and reaction kinetics
$Pt/CeO_2$ and $Pt_{AT}CeO_2$ were then evaluated for CO oxidation under $O_2$-rich (lean) conditions with a weight hourly space velocity (WHSV) of 300 L/g*h. The light-off curves and corresponding Arrhenius plots in Fig. 3a and Supplementary Fig. S8 show that $Pt/CeO_2$ is more active than $Pt_{AT}CeO_2$. For instance, $T_{50}$ (temperature for 50% CO conversion) for $Pt/CeO_2$ and $Pt_{AT}CeO_2$ are 180 and 335 °C, respectively. Five repeated light-off curves (Supplementary Fig. S9) display that two catalysts show stable catalytic performance and are both more active than the pristine $CeO_2$ (Supplementary Fig. S10). The obtained apparent activation energies ($E_a$) of $Pt/CeO_2$ and $Pt_{AT}CeO_2$ in the same temperature region (160–215 °C) by changing the WHSV are 44.6 and 82.4 kJ/mol (Fig. 3b), suggesting the reaction energy barrier in $Pt/CeO_2$ is distinctly lower than that in $Pt_{AT}CeO_2$. Moreover, the reaction order at ~200 °C in CO is ~0 for $Pt/CeO_2$ but +1 for $Pt_{AT}CeO_2$ (Fig. 3c). The near-zero reaction order in CO suggests the more favorable CO adsorption on $Pt/CeO_2$, which can be confirmed by a higher intensity of adsorbed CO peak in CO-DRIFTS (Fig. 2i) and more $CO_2$ evolution in temperature-programmed desorption of CO (CO-TPD, Supplementary Fig. S11). The kinetic feature of $Pt/CeO_2$ is also similar to that of reduced $Pt/CeO_2$ and $Pt_{AT}CeO_2$ samples obtained after a reduction in CO at 275 °C (Supplementary Fig. S12), as well as the Pt or Pd clusters on $CeO_2$[11,24]. This indicates that $Pt_1$ might sinter in $Pt/CeO_2$ under CO oxidation conditions. Increasing the surface coverage of $Pt_1$ is observed for two SACs with increasing CO partial pressure in CO-DRIFTS experiments at 100 °C (Supplementary Fig. S13a, b). However, the surface CO coverage in $Pt/CeO_2$ is higher than that in $Pt_{AT}CeO_2$ under the same condition, suggesting CO adsorption on $Pt/CeO_2$ is more kinetic-irrelevant, in agreement with the results in Fig. 3c. Moreover, the reaction orders in $O_2$ (Fig. 3d) over the two catalysts are also different, i.e., +0.3 for $Pt/CeO_2$ and -0 for $Pt_{AT}CeO_2$. Based on $O_2$-dependent CO-DRIFTS results (Supplementary Fig. S13c, d), higher $O_2$ partial pressure leads to a higher surface CO coverage on $Pt/CeO_2$; however, $O_2$ partial pressure does not have a noticeable effect on CO coverage on $Pt_{AT}CeO_2$. The above activity and kinetics suggest different dynamic behaviors of $Pt_1$ in $Pt/CeO_2$ and $Pt_{AT}CeO_2$ under CO oxidation condition. Both CO-DRIFTS (Fig. 2i) and CO oxidation kinetics suggest that two SACs hold different $Pt_1$ structures under CO oxidation condition.

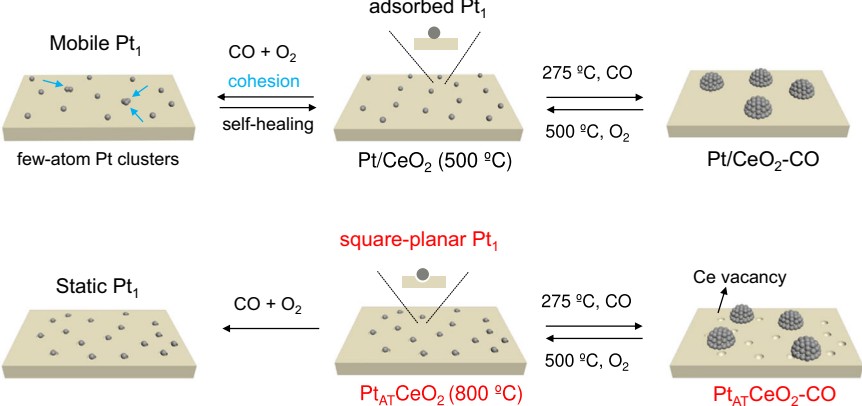

**Fig. 1 | Dynamics of $Pt_1$ under different conditions.** Schematic illustration of dynamic behaviors of single-atom $Pt_1$ in $Pt/CeO_2$ and $Pt_{AT}CeO_2$ under CO oxidation, reductive, and oxidative conditions at elevated temperatures.

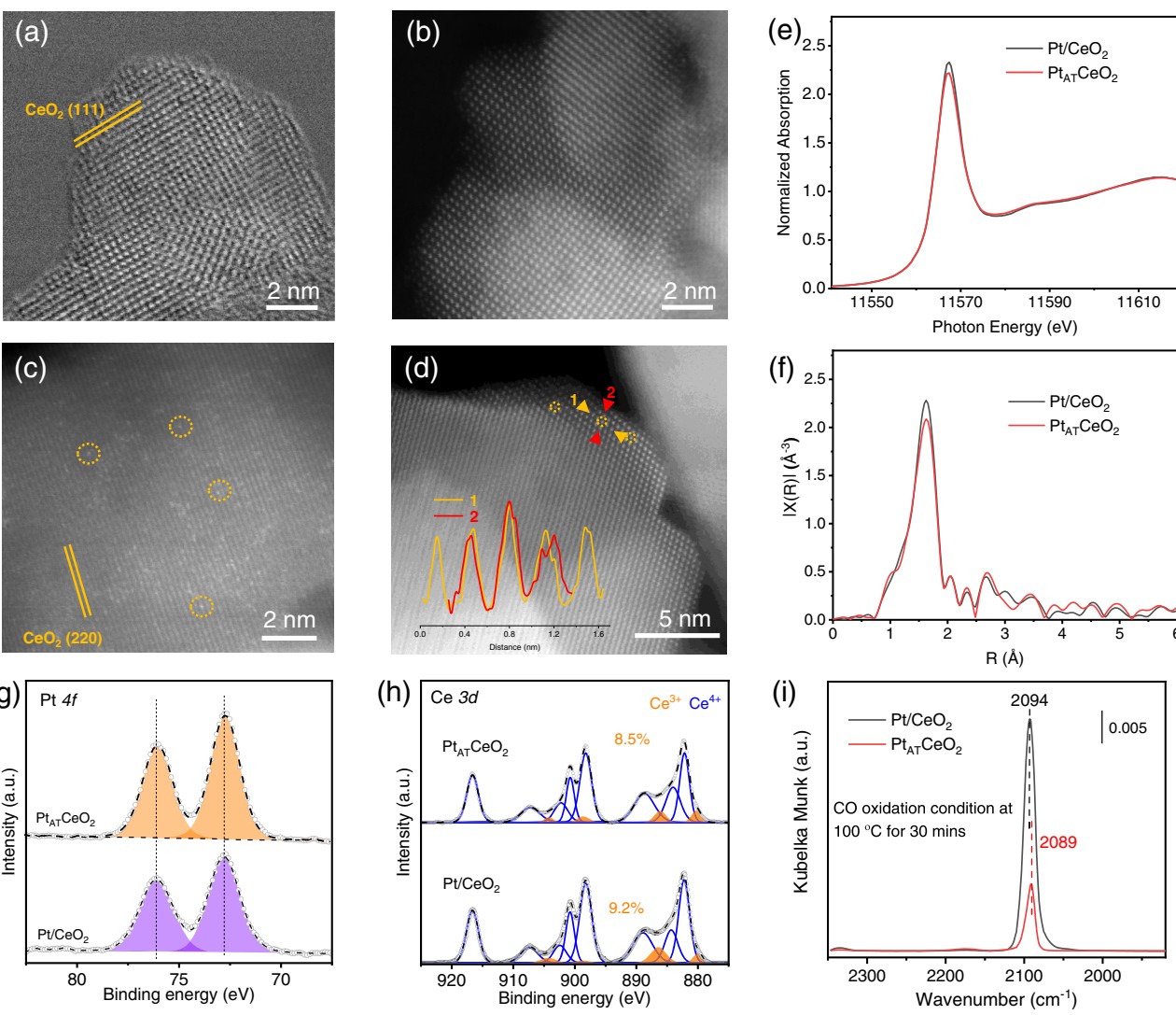

**Fig. 2 | Ex situ characterizations.** HAADF-STEM images of **a**, **b** Pt/CeO$_2$ and **c**, **d** Pt$_{AT}$CeO$_2$ (Pt$_1$ is marked as cycles, and line-scanning of a single Pt$_1$ is shown in the inset of (**d**). **e** Pt $L_3$-edge XANES and **f** the corresponding magnitude of the Fourier transform of the EXAFS spectra of Pt/CeO$_2$ and Pt$_{AT}$CeO$_2$. **g**, **h** Pt $4f$ and Ce $3d$ XPS spectra of Pt/CeO$_2$ and Pt$_{AT}$CeO$_2$. **i** In situ CO diffuse-reflectance infrared Fourier transform spectra (CO-DRIFTS) for Pt/CeO$_2$ and Pt$_{AT}$CeO$_2$ under CO oxidation condition at 100 °C for 30 min.

Additionally, Pt/CeO$_2$ and Pt$_{AT}$CeO$_2$ with a lower Pt loading (~0.1 wt%) also display a similar activity trend (Supplementary Figs. S14–S18). The detailed discussion is provided in Supplementary Information after Supplementary Figs. S14 and S15. It has been reported that surface reconstruction of CeO$_2$ at different calcination temperatures would affect the catalytic activity[25]. To minimize these effects, the CeO$_2$ support was pre-calcined at 800 °C for 10 h to yield 800CeO$_2$, followed by deposition of 0.1 wt% Pt (to maintain the atomically dispersed nature) at 500 and 800 °C to yield 0.1Pt/800CeO$_2$ and 0.1Pt$_{AT}$800CeO$_2$, respectively. Since the support was pre-calcined at 800 °C, these two samples exhibited similar porosity properties (Supplementary Fig. S17, Supplementary Table S3) and CeO$_2$ particle size (Supplementary Fig. S18) as the 800CeO$_2$ support. We found that the activity of 0.1Pt/800CeO$_2$ was still significantly higher than that of 0.1Pt$_{AT}$800CeO$_2$ (Supplementary Fig. S14), similar to Pt/CeO$_2$ and Pt$_{AT}$CeO$_2$ (Fig. 3a).

**Dynamic evolution under CO oxidation condition**

To probe the activity origin of Pt/CeO$_2$ and Pt$_{AT}$CeO$_2$, in situ CO-DRIFTS was first performed under CO oxidation conditions at different temperatures. At 35 or 80 °C, Pt$_{AT}$CeO$_2$ shows a similar weak IR peak

centered at ~2088 cm$^{-1}$ (Fig. 4b)[12,19], which becomes increasingly apparent at 120 °C. It has been reported that square planar Pt$_1$ hardly chemisorbs CO[26,27], and high-temperature treatment in CO + O$_2$ reconstructs new Pt$_1$ cations, which are capable of adsorbing CO[22], in agreement with our Pt$_{AT}$CeO$_2$ results. In contrast, Pt/CeO$_2$ shows a stronger adsorbed CO-Pt$_1$ peak (~2101 cm$^{-1}$) at 35 °C (Fig. 4a), which becomes more intense at 80 °C with a red shift, suggesting a possible reduction of Pt$_1$. An obvious shoulder (2000–2060 cm$^{-1}$) is observed in Pt/CeO$_2$ above 120 °C, along with an increased intensity of gaseous CO$_2$ in the IR cell, indicating that the reaction has begun and a new Pt species has formed. This shoulder is a typical characteristic of Pt clusters[10,11]. The above results show the reduction and sintering of Pt$_1$ in Pt/CeO$_2$ under the elevated reaction temperature. However, the features (<2000 cm$^{-1}$) ascribed to bridge adsorbed CO on the traditional large Pt NPs[28–30] are not observed, which can be seen in the reduced Pt/CeO$_2$ and Pt$_{AT}$CeO$_2$ (Supplementary Fig. S19). After cooling down to 35 °C in CO + O$_2$ from 250 °C, the Pt clusters feature can still be found (Fig. 4c). This feature disappears after cooling down in O$_2$, suggesting as-formed Pt clusters completely redisperse on CeO$_2$ in O$_2$. These suggest that Pt$_1$ in Pt/CeO$_2$ may only sinter into few-atom Pt clusters under reaction conditions due to the presence of self-healing of Pt clusters into Pt$_1$ under O$_2$-rich

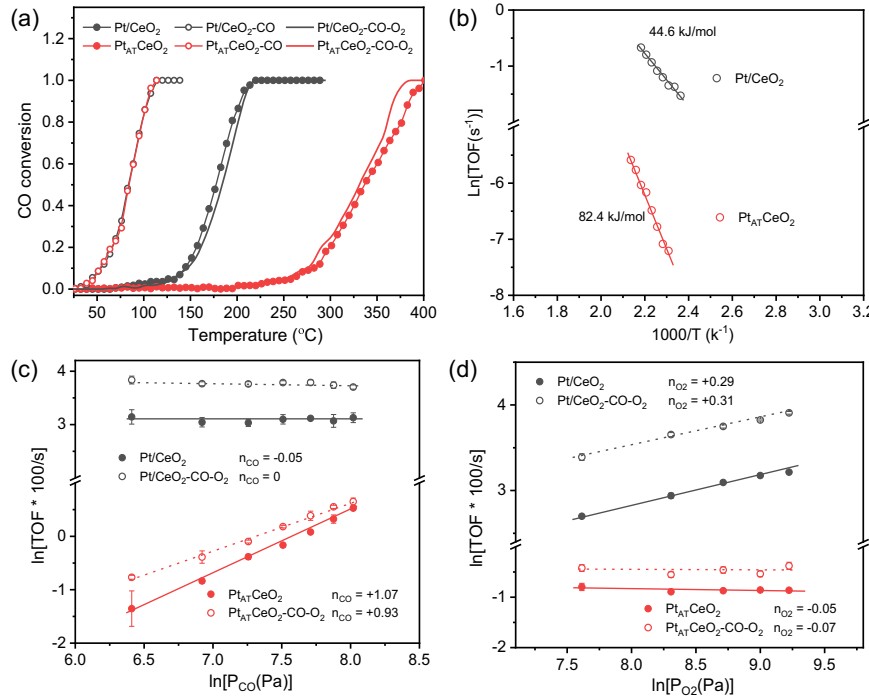

**Fig. 3 | Catalytic evaluation. a** CO oxidation performance (light-off curve) with 20 mg catalyst, and **b** Arrhenius plots of Pt/CeO$_2$ and Pt$_{AT}$CeO$_2$ with different catalyst loadings (4 mg Pt/CeO$_2$, 300 mg Pt$_{AT}$CeO$_2$). Reaction conditions: 1% CO and 4% O$_2$ balanced with Ar, catalyst diluted with SiC to 400 mg, total flow rate = 100 mL/min. Effect of **c** CO and **d** O$_2$ partial pressure on the reaction rate (TOF). Measurement conditions: P$_{CO}$ = 0.6–3 kPa, P$_{O2}$ = 4 kPa in (**c**); P$_{CO}$ = 1 kPa, P$_{O2}$ = 2–10 kPa in (**d**). The operating temperatures for Pt/CeO$_2$, Pt$_{AT}$CeO$_2$, Pt/CeO$_2$-CO-O$_2$, and Pt$_{AT}$CeO$_2$-CO-O$_2$ are 210, 210, 220, and 215 °C, respectively, in (**c**, **d**).

condition. The co-existence of Pt$_1$ cohesion and self-healing of Pt clusters in Pt/CeO$_2$ is crucial to maintain fully exposed Pt clusters under CO oxidation condition[31]. It should be emphasized that a gaseous CO$_2$ signal shows up at 35 °C and then disappears at 80 °C for Pt/CeO$_2$ in CO-DRIFTS (Fig. 4a). To validate this phenomenon, we then performed the temperature-programmed surface reaction (Supplementary Fig. S20). Once CO is introduced in O$_2$-treated samples at 35 °C, an immediate and ephemeral CO$_2$ evolution together with CO consumption is found only in Pt/CeO$_2$, suggesting the active O (or weakly bonded O) in Pt/CeO$_2$ can react with CO to form CO$_2$ at 35 °C. Meanwhile, a surface Pt$_1$ reconstruction in Pt/CeO$_2$ must occur due to the loss of surface O.

To further study the dynamic evolution of Pt$_1$, in situ XANES data were collected under CO oxidation condition. For Pt/CeO$_2$, an obvious decrease of the white line intensity is observed while switching the exposed atmosphere from ambient air to CO and O$_2$ at 25 °C (Fig. 4d), indicating Pt$_1$ transforms from near Pt$^{4+}$ in ambient air to -Pt$^{2+}$ in CO and O$_2$, as compared with the Pt reference (Supplementary Fig. S21). This finding explains why a CO$_2$ signal is observed at 35 °C in Pt/CeO$_2$ after introducing CO + O$_2$ (Fig. 4a, Supplementary Fig. S20). The decreased Pt valence can also be evidenced by the decreased first-shell Pt-O CN (5 to 3.1) from in situ EXAFS (Fig. 4f, Supplementary Table S2), clearly indicating the abovementioned active O in Pt/CeO$_2$ directly bonds with Pt$_1$. As further increasing temperature to 180 °C in CO + O$_2$, Pt valence in Pt/CeO$_2$ descends slowly (Fig. 4d). In contrast, white line intensity in Pt$_{AT}$CeO$_2$ is stable after flowing CO and O$_2$ at 25 °C or even at 100 °C, and it decreases only at 150 °C (Fig. 4e). The Pt-O CN decreases from 4.9 at 25 °C to 3.2 at 180 °C (Fig. 4g), indicating Pt$_1$ in Pt$_{AT}$CeO$_2$ reconstructs into lower-valence Pt$_1$ at increased temperature. Nonetheless, Pt-O CN in Pt$_{AT}$CeO$_2$ at 180 °C is still higher than 2.8 found in Pt/CeO$_2$ (Supplementary Table S2). After CO oxidation treatment at different temperatures, XPS data were collected quasi in situ. For Pt/CeO$_2$, the mild treatment at 30 °C does not influence the XPS signal, but a new feature appears at 180 °C, as seen in Fig. 4h. This suggests the formation of Pt species with the valence higher than 2.

Based on the previous studies[30], the oxidation of Pt NPs to PtO$_2$/PtO cluster mixture or the formation of thin PtO$_x$ oxide film can induce the formation of Pt cations (>2+). In comparison, this new Pt feature is not observed in Pt$_{AT}$CeO$_2$ under the same treatment condition (Fig. 4i). Therefore, we ascribe the newly formed Pt species under CO oxidation condition in Pt/CeO$_2$ as few-atom Pt clusters (Fig. 1). Moreover, the relatively lower Pt-O CN in Pt/CeO$_2$ at 180 °C is ascribed to the formation of few-atom Pt clusters under reaction condition by combining with CO-DRIFTS, XPS, and kinetics studies.

## Dynamic evolution under reductive-oxidative cycle and structural memory

To further investigate the difference between the two Pt$_1$ configurations, we designed a cohesion-redispersion cycle experiment. Two SACs are first treated in CO at 275 °C to form Pt/CeO$_2$-CO and Pt$_{AT}$CeO$_2$-CO. Pt NPs (1–2 nm in size) in reduced samples can be evidenced by HAADF-STEM images (Fig. 5a, d, Supplementary Fig. S22), XPS[32–34] (Supplementary Fig. S23), CO-DRIFTS (Supplementary Fig. S19), and Raman spectroscopy (Supplementary Fig. S24). Pt/CeO$_2$-CO and Pt$_{AT}$CeO$_2$-CO show similar enhanced CO oxidation reactivity (Fig. 3a) and similar reaction orders (Supplementary Fig. S12), ascribed to the presence of Pt clusters[13]. The percentage of Ce$^{3+}$ and surface defect-related O also increases after CO reduction (Fig. 5g, Supplementary Fig. S25). However, the increased activity is lost during the repeated CO oxidation experiments from 25 to 500 °C for both reduced catalysts (Supplementary Fig. S26). If we treat Pt/CeO$_2$-CO and Pt$_{AT}$CeO$_2$-CO in O$_2$ at 500 °C, both enhanced activities will also decrease and become similar to their respective initial activity (Fig. 5c, f). The activity loss is due to the redispersion of Pt NPs into Pt$_1$ evidenced by HAADF-STEM images (Fig. 5b, e, Supplementary Fig. S27) and CO-DRIFTS (Supplementary Figs. S28 and S29) results. Moreover, adsorbed CO-Pt$_1$ peak (Supplementary Figs. S28 and S29) in reoxidized Pt/CeO$_2$-CO-O$_2$ and Pt$_{AT}$CeO$_2$-CO-O$_2$ is located at -2095 and -2089 cm$^{-1}$, respectively, that is consistent with their

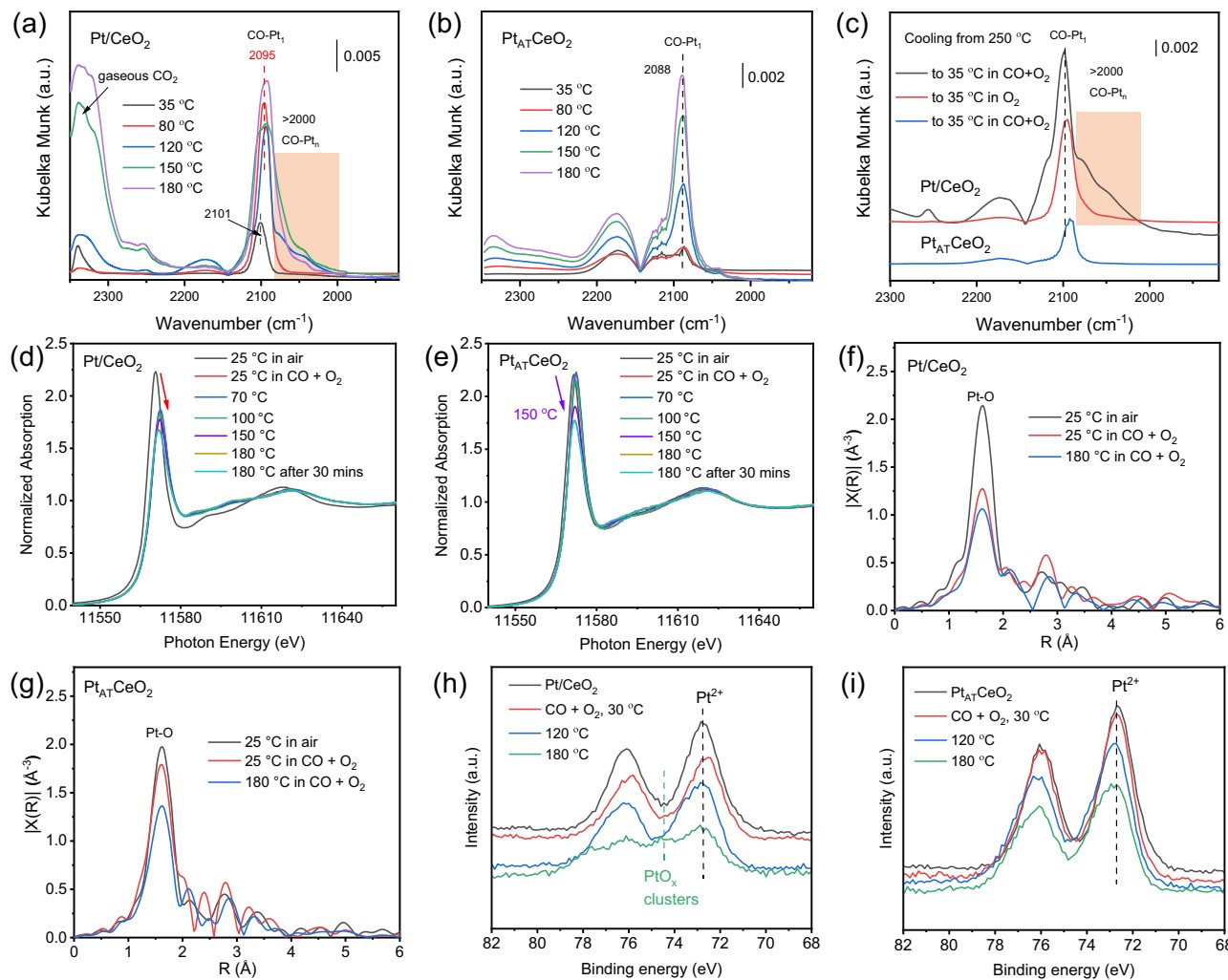

**Fig. 4 | In situ characterizations.** In situ CO-DRIFTS for **a** Pt/CeO$_2$ and **b** Pt$_{AT}$CeO$_2$ in CO and O$_2$ mixture as varying the temperature from 35 to 180 °C, as well as **c** the CO-DRIFTS after reaction at 250 °C, and cooling down to 35 °C in CO + O$_2$ or O$_2$. Pt $L_3$-edge in situ XANES of **d** Pt/CeO$_2$, and **e** Pt$_{AT}$CeO$_2$ at 25 °C in ambient air and different reaction temperatures (25, 70, 100, 150, 180 °C) in CO oxidation condition (CO/O$_2$ ratio is 1:4). the corresponding magnitude of the Fourier transform of the EXAFS of **f** Pt/CeO$_2$ and **g** Pt$_{AT}$CeO$_2$ at 25 °C in ambient air, and at 25 °C, 180 °C under reaction condition, k = 3–12.5 Å$^{-1}$ for the Fourier transform. Quasi in situ Pt $4f$ XPS spectra for **h** Pt/CeO$_2$ and **i** Pt$_{AT}$CeO$_2$ without or with treatment at different reaction temperatures (CO/O$_2$ ratio is 1:4) for 20 min. After treatment, gases were pumped for the XPS test.

respective fresh sample (Fig. 2i). This implies that two kinds of Pt$_1$ appear to have memory back to their initial state after a cohesion-redispersion cycle. More interestingly, the reaction kinetics also shows a similar memory behavior. Specifically, the reaction orders in CO for Pt/CeO$_2$ in three states (fresh-reduced-reoxidized) are all closer to 0 but change from 0.3 through −0.2 to 0.3 in O$_2$ (Fig. 5h). In Pt$_{AT}$CeO$_2$, the reaction order changes from 1.1 through 0 to 0.9 in CO, and from 0 through −0.2 to 0 in O$_2$. We also reduced Pt$_1$ in Pt/CeO$_2$ and Pt$_{AT}$CeO$_2$ with H$_2$ instead of CO, and the enhanced reactivity was also lost after a further reoxidation treatment at 500 °C (Supplementary Fig. S30). This indicates two catalysts have structural memory after both CO-O$_2$ and H$_2$-O$_2$ treatment cycles. Furthermore, T$_{50}$ of Pt/CeO$_2$ and Pt$_{AT}$CeO$_2$ after the sequential reductive-oxidative cycle (Fig. 5i) show that the cohesion-redispersion behavior of Pt$_1$ can be repeated many times. Therefore, we believe that after a reduction-reoxidation cycle, two Pt$_1$ configurations in Pt/CeO$_2$ and Pt$_{AT}$CeO$_2$ both return to their initial structure.

## Theoretical insight into the dynamic behaviors
To explain the above dynamic behaviors, the nascent Pt$_1$ structures of Pt/CeO$_2$ and Pt$_{AT}$CeO$_2$ are identified first. Based on the previous

studies[7,21–23], Pt$_{AT}$CeO$_2$ is dominated by square planar Pt$_1$ structure on CeO$_2$(111) terrace (Supplementary Fig. S31b) or step site (Supplementary Fig. S31c). However, Pt$_1$ configuration in Pt/CeO$_2$ is still unknown. To understand if Pt$_1$ in Pt/CeO$_2$ is another reported single-atom structure–adsorbed Pt$_1$ (Supplementary Fig. S31a)[35,36], we first compare EXAFS fitting results of Pt/CeO$_2$ and adsorbed Pt$_1$ models (Supplementary Fig. S31d), and the adsorbed PtO$_5$ model on CeO$_2$ (111) fits well with Pt/CeO$_2$. Then, we calculate the oxygen vacancy (V$_O$) formation energy of neighboring O of both adsorbed Pt$_1$ and square planar Pt$_1$. It is found that the V$_O$ formation energy of adsorbed Pt$_1$ is significantly lower than that of square planar Pt$_1$ (Supplementary Fig. S32). This indicates that neighboring O atoms of adsorbed Pt$_1$ are easier to remove, consistent with previous results (Fig. 4a, d, Supplementary Fig. S20), which further justifies our proposed adsorbed Pt$_1$ model for Pt/CeO$_2$. Therefore, we assume that our Pt/CeO$_2$ is mainly composed of adsorbed PtO$_5$ structure in air (Supplementary Fig. S31a). Under CO oxidation, the adsorbed PtO$_5$ adsorbs CO with the adsorption energy of −0.46 eV (Vi to Vii, Fig. 6a), but the adsorbed CO-PtO$_5$ is difficult to release CO$_2$ with an energy barrier of 1.5 eV (Vii to iV). Therefore, cycle 1 in Fig. 6a is unlikely to occur. Instead, the PtO$_5$ structure can easily transform into a PtO$_3$ structure (Vi to i) with an exothermic energy of

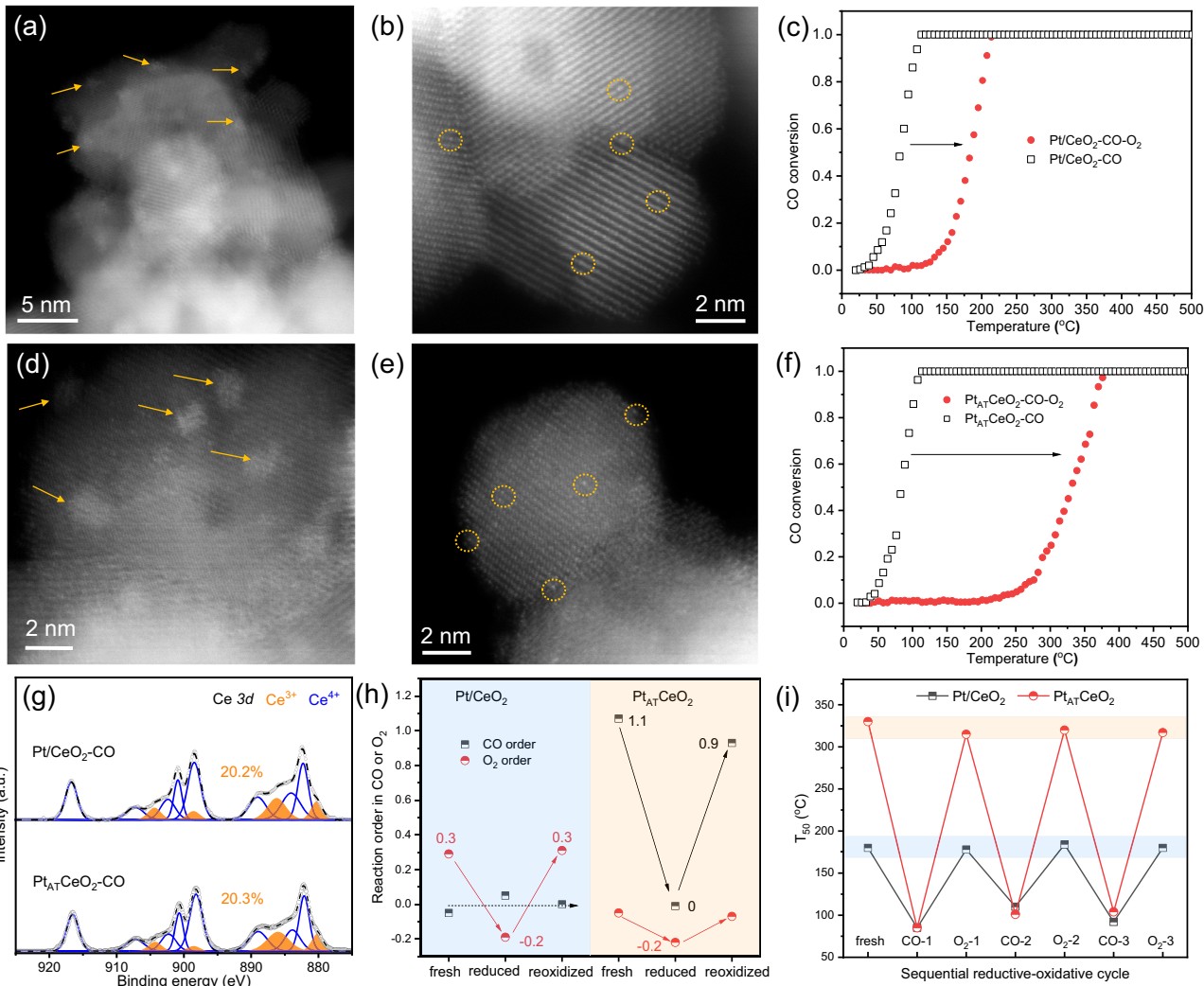

**Fig. 5 | Structural memory under reductive-oxidative cycle.** STEM images of **a** Pt/CeO₂-CO, **b** Pt/CeO₂-CO-O₂, **d** Pt$_{AT}$CeO₂-CO, and **e** Pt$_{AT}$CeO₂-CO-O₂. Light-off curves of reduced and reoxidized **c** Pt/CeO₂ and **f** Pt$_{AT}$CeO₂. **g** Ce *3d* XPS spectra of Pt/CeO₂-CO and Pt$_{AT}$CeO₂-CO; XPS data are collected after treatment without exposure to air. **h** Reaction orders in CO and O₂ for Pt/CeO₂ and Pt$_{AT}$CeO₂ in three states. **i** $T_{50}$ results of Pt/CeO₂ and Pt$_{AT}$CeO₂ after a sequential reductive-oxidative cycle. Reaction conditions in (**c, f, i**) are the same as that in Fig. 3a.

−0.3 eV, consistent with in situ XAS result (Fig. 4d). The formed PtO₃ adsorbs CO strongly (i to ii), then releases CO₂ to form PtO₂ with an energy barrier of 0.69 eV (ii to iii). O₂ fills the O$_v$ around PtO₂ to form PtO₄ with an exothermic energy of −1.52 eV, followed by a CO adsorption (iii to iV to V). The adsorbed CO-PtO₄ loses CO₂ to form PtO₃ with the energy barrier of 0.5 eV to complete cycle 2. PtO₂ can also adsorb CO strongly to form CO-PtO₂; however, Pt-O scission occurs spontaneously with a strong exothermic energy of −3.7 eV (Viii to iX) to form CO-PtO structure (iX). Assuming that there are two CO-PtO on the CeO₂ surface, the calculated Pt-Pt cohesion energy barrier is 0.8 eV (iX to X), which indicates the possible Pt-Pt cohesion under reaction condition in Pt/CeO₂.

In Fig. 6b, square planar Pt₁ on CeO₂(111) terrace is adpoted[23]. First, CO adsorbs on square planar Pt₁ (PtO₆) with an adsorption energy of −0.6 eV (i to ii), much lower than −2.21 eV observed on PtO₃ in Pt/CeO₂. This is consistent with the stronger IR signal found in Pt/CeO₂ at 35 and 80 °C (Fig. 4a, b). Thereafter, CO-PtO₆ requires a moderate energy barrier of 0.5 eV to release CO₂ to form PtO₅ (ii to iii), which is why the white line intensity of Pt$_{AT}$CeO₂ only decreases at 150 °C (Fig. 4e). PtO₅ can either adsorb O₂ or CO to form PtO₅(O₂) (iii to iV) or CO-PtO₅ (iii to Vi). However, CO adsorbed on PtO₅(O₂) requires an energy barrier of 1.12 eV to release CO₂ (V to i), which

makes cycle 1 less favorable. In contrast, CO-PtO₅ loses CO₂ to form PtO₄ with an energy barrier of 0.81 eV. It should be noted that the formed PtO₄ here has a similar structure as square planar Pt₁ on the CeO₂ step site (Vii), so the step-site situation is not considered individually. PtO₄ can either adsorb O₂ to close cycle 2 or adsorb CO to form CO-PtO₄, which will further transform to PtO₃ after CO₂ removal. PtO₃ then adsorbs CO strongly, but adsorbed CO-PtO₃ is unlikely to transform to PtO₂ due to its endothermic nature. Instead, it adsorbs O₂ with an adsorption energy of −2.09 eV to complete cycle 3, which prevents the sintering of Pt₁. The overall energy barriers of relatively favorable cycle 2 in adsorbed Pt₁ in Pt/CeO₂ and square planar Pt₁ in Pt$_{AT}$CeO₂ are 0.69 and 0.81 eV, respectively. Such a small difference should not induce the huge activity difference in Fig. 3, which also implies parts of Pt₁ in Pt/CeO₂ sinter under CO oxidation condition. Supplementary Fig. S33 shows the simulated CO vibrational frequencies on both adsorbed and square planar Pt₁. The calculated vibrational frequencies of adsorbed CO on PtO₃ for both Pt₁ structures are consistent with the CO-DRIFTS results, indicating that the observed CO-Pt₁ band in CO-DRIFTS can be attributed to the adsorbed CO on PtO₃. What sets Pt/CeO₂ apart is that PtO₃ further transforms into PtO and then sinters at higher temperatures. However, PtO₃ is relatively stable in Pt$_{AT}$CeO₂.

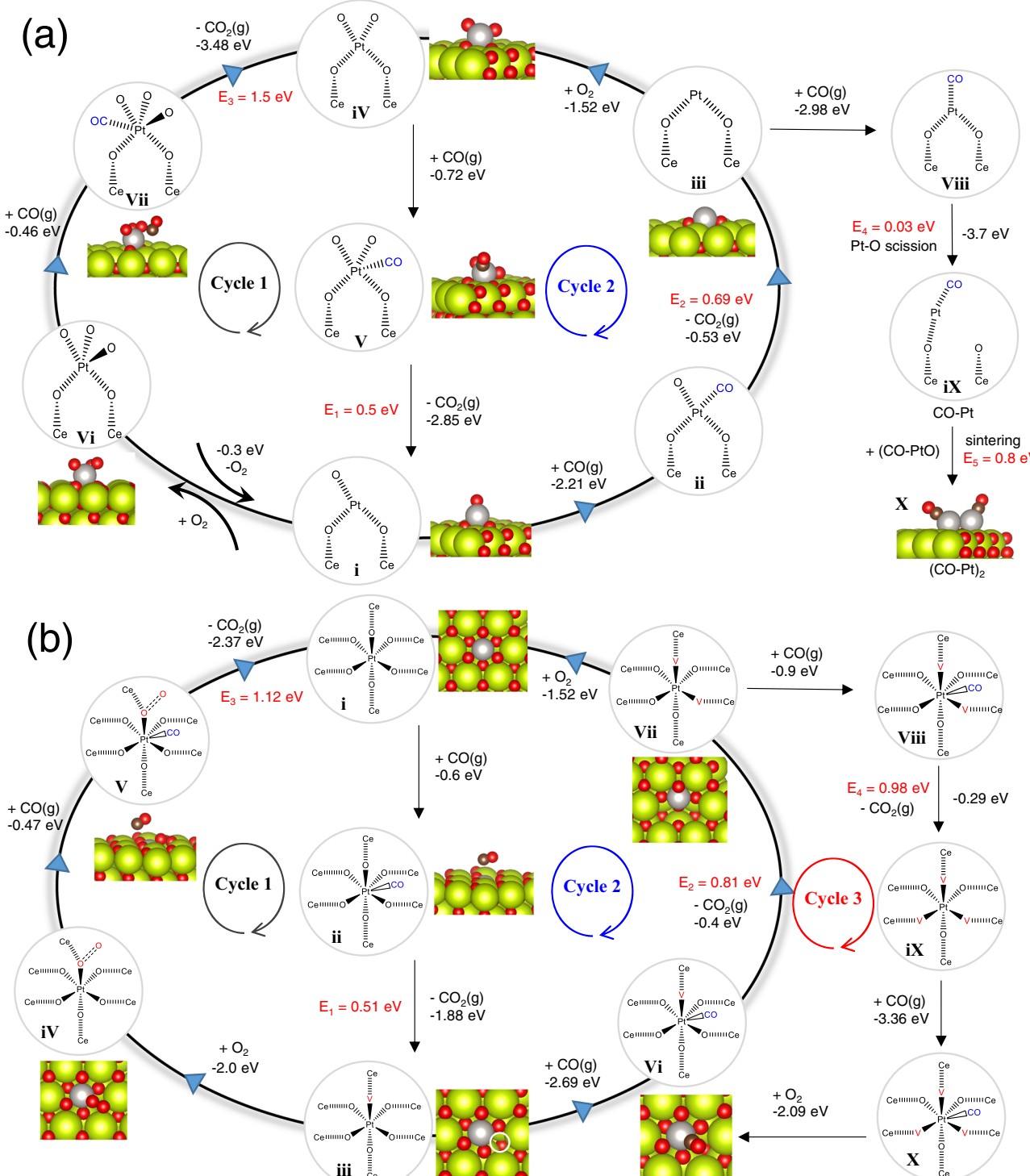

**Fig. 6 | DFT calculation.** CO oxidation reaction mechanism on **a** adsorbed $Pt_1$ and **b** square planar $Pt_1$ on $CeO_2$ (111) terrace. The main model structures are shown in the reaction cycle.

It has been reported that Ce vacancy ($V_{Ce}$) is more difficult to generate compared to the $V_O$[37]. However, we can speculate that the exsolution of square planar $Pt_1$ on the $CeO_2$ terrace in $Pt_{AT}CeO_2$ is a strategy to generate $V_{Ce}$. The possible reason for the formation of square planar $Pt_1$ only at 800 °C is the lattice expansion of $CeO_2$ at high temperatures, as seen in in-situ XRD (Supplementary Fig. S34) and the favorable migration of cerium cations to PGMs surface at higher temperatures in $O_2$[38–40]. These could facilitate the surface $CeO_2$ reconstruction around the Pt atom or migration of cerium cations onto

the Pt surface to form a local square planar $Pt_1$ structure (Supplementary Fig. S35). We then construct two models for $Pt/CeO_2$-CO and $Pt_{AT}CeO_2$-CO with a five-atom Pt NP on $CeO_2$ (111) surface with and without $V_{Ce}$ (Supplementary Fig. S36) to simulate their redispersion process. This process includes the oxidation of the five-atom Pt NP to the $PtO_x$ cluster and the redispersion of the top Pt atom (Supplementary Fig. S37). The results indicate that the redispersion of the top Pt atom into a surrounding $V_{Ce}$ is energetically more favorable than the intact $CeO_2$ surface (formation energy: −2.02 eV vs 0.26 eV). Therefore,

from a thermodynamic point of view, $V_{Ce}$ generated after the exsolution of square planar $Pt_1$ could, in return, trap Pt atoms more readily during a reoxidation treatment to form a square planar structure again. In contrast, Pt clusters are formed without surface $V_{Ce}$ after reducing the adsorbed $Pt_1$ on $CeO_2$, and then it redisperses into adsorbed form after a reoxidation treatment. This explains why two SACs have the memory to return to their respective native structure. The different dynamic evolution under CO oxidation condition and their structural memory behaviors under reductive-oxiditive treatment cycle is due to their various initial $Pt_1$ location on $CeO_2$ driven by different calcination temperatures. Therefore, designing SACs with tunable location is important to maximize their catalytic performance in the future.

In summary, $Pt/CeO_2$ and $Pt_{AT}CeO_2$ were fabricated via two different annealing temperatures of 500 and 800 °C. Pt atoms are both atomically dispersed in nascent $Pt/CeO_2$ and $Pt_{AT}CeO_2$, evidenced by the combined characterization results. These two catalysts display dramatically different catalytic activity toward CO oxidation and different apparent activation energies and reaction orders in CO and $O_2$. These differences could be explained by the different initial $Pt_1$ local configurations, where $Pt_1$ in $Pt/CeO_2$ and $Pt_{AT}CeO_2$ are dominated by adsorbed $Pt_1$ and square planar $Pt_1$, respectively. Under reaction condition, adsorbed $Pt_1$ in $Pt/CeO_2$ sinters into few-atom Pt clusters; however, square planar $Pt_1$ in $Pt_{AT}CeO_2$ is strongly anchored to the support with a decrease in the Pt-O coordination number. After the treatment in CO at 275 °C, both types of $Pt_1$ transform to Pt NPs, which inevitably redisperse at the elevated temperature in $O_2$ or even under $O_2$-rich reaction condition. What is more interesting is that the initial thermal treatment creates memory on the support where the Pt atoms return under CO oxidation or oxidative conditions, potentially providing a catalyst self-healing after severe catalyst sintering.

## Methods

### Synthesis of $Pt/CeO_2$ and $Pt_{AT}CeO_2$

$CeO_2$ powder was synthesized by the precipitation method with ammonia, followed by washing with DI water, drying, and calcination at 500 °C in air for 4 h. Tetraammineplatinum(II) nitrate was then impregnated on $CeO_2$ powder by the incipient wetness impregnation, with the calculated Pt weight loadings of 1%. After impregnation, the samples were dried at 100 °C for 12 h, followed by calcination at 500 °C and 800 °C in air for 10 h to yield $Pt/CeO_2$ and $Pt_{AT}CeO_2$ catalysts, respectively. $Pt/CeO_2$-CO and $Pt_{AT}CeO_2$-CO were obtained after treating the fresh samples in CO/Ar (20 mL/min) at 275 °C for 20 min. $Pt/CeO_2$-CO-$O_2$ and $Pt_{AT}CeO_2$-CO-$O_2$ were achieved after further treating the reduced samples in air at 500 °C for 10 h. The low-loading catalysts were synthesized by the same method.

### Activity measurements

CO oxidation experiments of fresh $Pt/CeO_2$ and $Pt_{AT}CeO_2$ were carried out in a fixed-bed flow reactor. Then, 20 mg of catalyst sieved between 40 and 80 mesh was diluted with 380 mg washed SiC powder and then loaded together into the reactor tube. The reaction temperature was ramped up from 20 to 500 °C with a heating rate of 3 °C /min in the mixture of 1 mL/min CO, 4 mL/min $O_2$, and 95 mL/min Ar, with a weight hourly space velocity (WHSV) of 300 L/g*h. The reactor was cooled down to 20 °C in the above reaction mixture for the next light-off test. The product concentration was measured by a gas chromatograph Agilent 3000 Micro GC. The activity measurements of reduced and reoxidized catalysts were performed under the same reaction condition after the in situ pretreatment in the same fixed-bed flow reactor. CO oxidation kinetic measurements were carried out under different reaction conditions by controlling the CO conversion lower than 8%. The partial pressures of CO and $O_2$ were adjusted by changing their flow rates. To study the effect of CO partial pressure on reaction rate, the partial pressure of $O_2$ was kept at 4 kPa, and the partial pressure of CO changed between 0.6 and 3 kPa. To study the effect of $O_2$ partial

pressure on reaction rate, the partial pressure of CO was kept at 1 kPa, and the partial pressure of $O_2$ changed between 1 and 10 kPa. The reported reaction rates were normalized by the total numbers of Pt, assuming that all Pt are accessible.

### Characterization

Powder X-ray diffraction (XRD) patterns were collected using a Rigaku Miniflex 600 equipped with Cu Kα radiation, with an operating voltage of 40 kV and a current of 15 mA. All samples are collected from 15 to 65° with a speed of 0.5°/min. In situ XRD was carried out in an XRD cell on a PANalytical Empyrean X-ray diffractometer equipped with Cu Kα radiation, with an operating voltage of 45 kV and a current of 40 mA. Quasi in situ X-ray photoelectron spectroscopy measurements were carried out with a Physical Electronics Quantera SXM Scanning X-ray Microprobe with a focused monochromatic Al Kα X-ray (1486.7 eV) source and multi-channel detector. Prior to the test, the samples were pretreated in a preparation chamber under different temperatures and gases, i.e., 180 °C in CO and $O_2$ or 275 °C in CO. After the pretreatment, the samples were directly transferred into the XPS detection chamber for the test without exposure to other gases. All spectra, including Pt, Ce, and O in binding energies, were charge corrected by shifting the $Ce^{4+}$ $3d_{5/2}$ line to 916.7 eV[41]. Diffuse-reflectance infrared Fourier transform spectroscopy with CO as the probe molecule (CO-DRIFTS) was carried out on a Thermo Scientific IS-50R FTIR with the MCT/A detector. Prior to analysis, approximately 40 mg of the sample was pretreated at 200 °C for 30 min with $O_2$/He flow in a DRIFTS reaction chamber. A spectral resolution of 4 $cm^{-1}$ was used to collect spectra, and each spectrum in the work is an average of 32 scans. Ex situ XAS measurements were performed at X-ray Science Division bending-magnet beamline at sector 20 of the Advanced Photon Source operating at Argonne National Laboratory. In brief, the samples after calcination were pressed and covered into thin sheets in air before the test. In situ XAS measurements were carried out at the Stanford Synchrotron Radiation Light Source (SSRL) at beamline 9-3 in fluorescence mode. The catalysts were characterized by in situ XAS at the Pt $L_3$-edge (11564 eV) using an in-house built cell with a 4-mm ID glassy carbon tube. The catalyst and standard samples were scanned simultaneously in transmission and fluorescence detection modes using ion chambers and a 100-element solid-state Ge monolith detector (Canberra). XANES and EXAFS data processing and analysis were performed using Athena and Artemis programs of the Demeter data analysis package[42,43]. The detailed measurement and analysis methods can be seen in our previous study[21]. The theoretical EXAFS signals for the Pt-O path of $Pt_1$ adsorbed on $CeO_2$ were generated using the FEFF6 code from a Pt doped on the $CeO_2$ model. The theoretical EXAFS signals were fitted to the data in R-space using Artemis by varying the coordination numbers of the single scattering paths, the effective scattering lengths, the bond length disorder, and the correction to the threshold energy, $\Delta E_0$ (common for all paths since they are all from the same FEFF calculation). $S_0^2$ (the passive electron reduction factor) was obtained by first analyzing the spectrum for the Pt oxide, and the best-fit value (0.90) was fixed in the fit. The k-range used for fitting was 3–14 $Å^{-1}$ while the R-range was 1.2–2 Å for the model that only includes the Pt-O scattering shell. High-angle annular dark-field scanning transmission electron microscopy (HAAD-STEM) images were collected on a Nion UltraSTEM microscope operated at 100 keV. Inductively coupled plasma-atomic emission spectroscopy (ICP-AES) was performed using an Optima 2100 DV spectrometer (PerkinElmer Corporation). $N_2$ adsorption-desorption isotherms were analyzed at 77 K on the Micromeritics gas adsorption apparatus (Quadrasorb-EVO, Quantachrome Corporation, America). Prior to analysis, all samples were pretreated at 200 °C for 4 h in a vacuum condition. The specific surface area was calculated using the Brunauer−Emmett−Teller (BET) equation. Temperature-programmed desorption of CO was performed on Micromeritics Autochem 2920 with a TCD detector and coupled mass spectroscopy

(MS) detector. Prior to analysis, the sample was pretreated at 500 °C in He for 30 min, followed by cooling down to room temperature in He. The treated sample was then exposed to 10% CO/Ar before ramping in He. Temperature-programmed surface reaction (TPSR) was carried out with the same instrument. The visible Raman spectra (532 nm) were collected on a Horiba LabRAM HR Raman/FTIR microscope equipped with a Synapse Charge Coupled Device (CCD) camera and an in situ sample cell (Linkam CCR 1000). All Raman spectra were conducted at room temperature, including the one after CO pretreatment at 275 °C. No obvious changes upon extended laser exposure were observed in the sample.

## DFT calculations

The periodic density function theory (DFT) calculations were carried out with the CP2K package[44]. The generalized-gradient approximation (GGA) with Perdew−Burke−Ernzerhof (PBE) functional was used to evaluate the exchange and correlation[45]. The wave functions were expanded in a molecularly optimized double-Gaussian basis set, with an auxiliary plane wave basis set with a cutoff energy of 500 Rydberg. The scalar relativistic norm-conserving pseudo-potentials were employed to model the core electrons[46] with 18, 12, and 6 valence electrons for Pt, Ce, and O, respectively. The only Γ-point in the reciprocal space mesh was used for integrating the Brillouin zone. The DFT + U method[47], based on the Mullikan 4f state population analysis, was used to describe the Ce 4f electrons. A U value was set at ~4.1 eV in line with the previous literature[48], which ensures that the redox property is reproduced correctly[49]. Grimme's third-generation DFT-D3 approach was used to describe dispersion corrections[50]. The $CeO_2(111)$ surfaces were used to model the $CeO_2$ substrate, constructed with cell dimensions of $15.344 \times 13.288 \times 27.529$ Å with 15-Å vacuum space to minimize the interaction between slabs. Geometry optimization was performed based on the BFGS method. The convergence criterion used for geometry optimizations was a maximum force of $0.01\,eV\,Å^{-1}$. Spin polarization was considered in all calculations.

## Data availability

The data generated in this study are provided in the Supplementary Information. More detailed data that support the findings of this study are available from the corresponding author upon reasonable request.

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

## Acknowledgements

This work was supported by the U.S. Department of Energy (DOE), Office of Basic Energy Sciences (SC), Division of Chemical Sciences (grant DE-FG02-05ER15712). We also acknowledge the U.S. Department of Energy (DOE) Energy Efficiency and Renewable Energy, Vehicle Technologies Office, for the support to Z.Z. and J.T. Use of the Stanford Synchrotron Radiation Lightsource, SLAC National Accelerator Laboratory, is supported by the U.S. Department of Energy, Office of Science, Office of Basic Energy Sciences. Co-ACCESS is supported by the U.S. Department of Energy, Office of Basic Energy Sciences, Chemical Sciences, Geosciences and Biosciences Division. A part of the research described in this paper was performed in the Environmental Molecular Sciences Laboratory (EMSL), a national scientific user facility sponsored by the DOE's Office of Biological and Environmental Research and located at PNNL.We acknowledge the use of facilities within the Eyring Materials Center at Arizona State University supported in part by NNCI-ECCS-1542160.

## Author contributions

Z.Z. and Y.W. conceived and planned the research. J.T. performed DFT computations. S.Y. carried out TEM measurements. S.R.B., J.H., and A.S.H. aided in the XAS experimental design and data collection. Y.B.L. performed the XAS modeling and CO-DRIFTS measurements. Y.X.L. and W.H. performed XRD measurements. D.J. performed Raman measurements. M.H.E. performed XPS measurements. Z.Z. synthesized the catalysts and performed other experimental and analytical studies. Z.Z., A.K.D., and Y.W. wrote the paper. All authors discussed the results and commented on the paper.

## Competing interests

The authors declare no competing interests.
