## [Peer Review File · Nature Communications]

Memory-dictated Dynamics of Single-atom Pt on CeO₂ for CO OxidationReviewers' comments:

Reviewer #1 (Remarks to the Author):

The manuscript by Zhang et al. reports two ceria-supported single atom Pt catalysts with different coordination environments, Pt/CeO₂ (adsorbed Pt₁) and Pt_{at}/CeO₂ (square planar Pt₁ embedded in the ceria surface). The structural evolution of these two SACs was investigated by a multitechnique approach including various spectroscopic and microscopic methods in conjunction with DFT calculations. The authors found that both SACs show structural memory behaviors under reductive-oxidative treatment cycles. It is an interesting work on single atom catalysis. However, I cannot recommend this manuscript for publication in its current form.

- 1) The assignment and deconvolution of the Ce³⁺ and Ce⁴⁺ peaks in the XPS spectra (Fig. 1h and Fig. 4g) are not correct. The Ce XPS data should be re-analyzed based on a proper assignment and deconvolution process.
- 2) I did not see a significant difference in frequency between the IR signals (2094 vs. 2089 cm⁻¹, Fig. 1i) of CO adsorbed on two different types of Pt single atoms in Pt/CeO₂ and Pt_{at}/CeO₂.
- 3) The dynamic structural evolution of Pt from charged single atoms to reduced clusters was tracked by CO vibrations that vary in frequency depending on both oxidation states and coordination numbers. The formation of Pt clusters was characterized by low-lying CO bands around 2070 and 2050 cm⁻¹ (Fig. S19). However, in the previous work (Ref. 22) they assigned these CO bands to reduced Pt₁. This is a key point and should be discussed in more detail based on the recent study of structure changes of Pt/CeO₂, where IR vibrational bands were observed for CO molecules adsorbed on surface sites exposed by both Pt and ceria support (modified via interaction with Pt, see J. Phys. Chem. C 2022, 126, 9051).
- 4) The IR band at 2114 cm⁻¹ was assigned to CO bound to oxidized PtO_x clusters. However, a closer analysis shows that the 2114 cm⁻¹ signal is proportional in intensity to the one at about 2170 cm⁻¹ (Fig. 3a-c), revealing that they are more likely to originate from the CO gas phase contribution, rather than CO-PtO_x.
- 5) The redispersion of Pt NPs into Pt₁ in O₂ at 500 °C was visualized by HAADF-STEM. It would be better to mark them in the corresponding images (Fig. S27).
- 6) I would suggest to perform additional calculations on the CO vibrational frequency. The agreement of computed and experimental frequencies of CO adsorbed on different Pt₁ sites would strengthen the conclusions, in particular the proposed structure of adsorbed Pt₁ on ceria.
- 7) It has been reported that differently shaped ceria NPs could undergo massive surface restructuring under activation and reaction conditions at elevated temperatures. The morphological changes of ceria support should have significant impact on the interaction with Pt and the activity for CO oxidation (see e.g. J. Phys. Chem. Lett. 2020, 11, 7925).

Reviewer #2 (Remarks to the Author):

This paper reports the dynamic properties of Pt/CeO₂ single atom catalysts according to their initial

calcination step (500°C for Pt/CeO₂ and 800°C for PtATCeO₂). The Dynamic evolution of the Pt nanostructure was probed under CO oxidation condition and redox sequences by advanced in-situ characterization techniques (EXAFS, DRIFT, RAMAN) and DFT calculations. Noteworthy results deal with the initial Pt nanostructure (Pt single atoms interaction with ceria surface) which was found to depend on the initial annealing temperature. Pt nanoparticles formed by a reduction step redisperse under oxidizing conditions and return to their initial nanostructure. However, these main findings are not original as dynamic properties of Pt/CeO₂ are well documented in the literature including the initial nanostructure of Pt/CeO₂ single atoms as a function of the annealing temperature. The location of Pt single atoms on ceria after a calcination at low (400-500°C) and high (750-800°C) has been already fully described by similar characterization techniques and DFT (see for instance : [10.1002/anie.201402342](https://doi.org/10.1002/anie.201402342), <https://doi.org/10.1038/s41929-020-00508-7>, <https://doi.org/10.1021/acs.jpcc.2c02420>, <https://doi.org/10.1016/j.apcatb.2020.119304>). The reversibility between single atoms and nanoparticles as a function of temperature and redox atmosphere and its consequence on CO oxidation catalytic activity is also well reported in previous papers (see for instance : [10.1002/anie.200803126](https://doi.org/10.1002/anie.200803126), [10.1039/d0cy00732c](https://doi.org/10.1039/d0cy00732c)).

One of the main difficulties with Pt/CeO₂ catalysts is to differentiate oxidized Pt clusters and Pt single atoms using STEM or EXAFS. The presence of few oxidized Pt clusters can strongly modify the surface reactivity of the catalyst. From my point of view, it is difficult to clearly conclude from the results of the characterization techniques used in this study that the investigated catalysts only contain adsorbed single atoms and no oxidized Pt clusters. Furthermore, the Pt-Pt coordination numbers determined by EXAFS are not reported. In a recent study ([10.1021/jacs.9b09156](https://doi.org/10.1021/jacs.9b09156)), Resasco et al. have demonstrated that isolated Pt atoms cannot chemisorb CO at low temperature, contrary to the catalysts of this study (Fig 1i). The same paper and another recent study (<https://doi.org/10.1021/acs.jpcc.1c06784>) have assigned bond CO with a stretch at 2095 cm⁻¹ to adsorption on oxidized Pt clusters. Similar IR bands were observed in this study at 2094 cm⁻¹ for Pt/CeO₂ and to a lesser extent at 2089 cm⁻¹ for PtATCeO₂ at 100°C under CO oxidation condition but were attributed to ionic Pt although they could indicate the presence of small oxidized Pt clusters, especially on the catalyst calcined at 500°C. In addition, Fig. S13a clearly shows a small band near 2072 cm⁻¹, characteristic of Pt clusters as observed after CO reduction (Fig S19). Another insight of the possible presence of PtO_x clusters in Pt/CeO₂ is its activation energy for CO oxidation which was found to be 44.6 kJ/mole. This value corresponds to reduced catalysts containing Pt nanoparticles (see for instance [10.1021/jacs.9b09156](https://doi.org/10.1021/jacs.9b09156), [10.1039/d0cy00732c](https://doi.org/10.1039/d0cy00732c)). The presence of small PtO_x clusters in Pt/CeO₂ could also explain its good catalytic activity (Fig. 2a).

According to the authors, the Pt single atoms in Pt/CeO₂ would be easier to sinter and to form Pt nanoparticles in reducing atmosphere than ones in PtATCeO₂. If so, why the two catalysts achieve the same activity for CO oxidation after the same reduction step in CO ? By the same way, Fig. S26 seems to indicate that the redispersion of Pt is slower on PtAT/CeO₂. Is-it coherent with the proposed mechanism ?

I would recommend before considering this paper for publication to clearly confirm the absence of any PtOx clusters on Pt/CeO₂ by CO pulse chemisorption and H₂-TPR. This latter technique could give an insight on the presence of PtOx clusters from the onset reduction temperature as isolated Pt²⁺ sites are inactive towards H₂ dissociation (10.1021/acscatal.8b00330, 10.1039/c6cp00627b, <https://doi.org/10.1021/acscatal.1c04565>). According to J. Resasco et al. (10.1021/jacs.9b09156), catalysts in which Pt is reduced at low temperatures (around 200 °C) contain PtOx clusters.

Reviewer #3 (Remarks to the Author):

This manuscript investigates the reversible structural evolution of Pt/CeO₂ and PtATCeO₂ SACs under different atmospheres. In addition, the two SACs catalysts display dramatically different catalytic activity towards CO oxidation, which are explained by the different initial Pt¹ local configurations. However, considering some catalytic behavior of the SACs, the identification of the active site and the related structural-activity relationship is suspicious, which is needed to be addressed in details.

(1) P13, Line 310: The processing temperatures for Pt/CeO₂-CO and Pt/CeO₂-CO-O₂ are not shown in the Methods section.

(2) For PtATCeO₂ sample treated at 800 °C, the particle size of CeO₂ increases from 12 to ~22 nm with the significantly decreased surface area. In addition to the surface reconstruction, the agglomeration of bulk phase CeO₂ is inevitable. If it is possible that a large amount of Pt enters the lattice of CeO₂? This will affect the amount of available Pt atoms on the surface.

(3) In Fig. 1i, considering the spectral resolution (4 cm⁻¹), the different IR band is too small (5 cm⁻¹). In addition, why the CO-DRIFT spectra were collected at 100°C under reaction conditions? The CO atmosphere and room temperature should be better for the characterization.

(4) According to the evaluation results, the Pt NPs show significantly better catalytic activity than SACs. Therefore, confirming the real active site in the two fresh SACs is very important for the following structure-activity relationship. As shown in Fig. 2a and Fig. S14, Pt/CeO₂ and 0.1Pt/CeO₂ show significant different light-off temperatures (100 vs. 200 °C). At 200 °C, the CO conversion is 100% over Pt/CeO₂. If the active sites are the same for these two SACs catalysts, the CO conversion should be larger than 10% for 0.1Pt/CeO₂, which is accurately ~2%. This indicates that the active site in Pt/CeO₂ is more active. How does the author comment on this result?

(5) Similar results are shown over PtATCeO₂ and 0.1PtATCeO₂ (Figure 2a and S14), the light-off temperatures are 200 and 300 °C, respectively. In addition, the catalytic activity of 0.1PtATCeO₂ is

similar with that over CeO₂ (Figure S10). Therefore, the Pt cluster and CeO₂ are important active sites for the fresh Pt/CeO₂ and PtATCeO₂ catalysts, which are not well considered in this manuscript.

Reviewer #1: The manuscript by Zhang et al. reports two ceria-supported single atom Pt catalysts with different coordination environments, Pt/CeO₂ (adsorbed Pt₁) and Pt_{at}CeO₂ (square planar Pt₁ embedded in the ceria surface). The structural evolution of these two SACs was investigated by a multitechnique approach including various spectroscopic and microscopic methods in conjunction with DFT calculations. The authors found that both SACs show structural memory behaviors under reductive-oxidative treatment cycles. It is an interesting work on single atom catalysis. However, I cannot recommend this manuscript for publication in its current form.

1) The assignment and deconvolution of the Ce³⁺ and Ce⁴⁺ peaks in the XPS spectra (Fig. 1h and Fig. 4g) are not correct. The Ce XPS data should be re-analyzed based on a proper assignment and deconvolution process.

Reply: We appreciate the constructive comments provided by the reviewer. We acknowledge that XPS peak fitting of Ce 3d spectra from CeO_x is a complex and highly debated topic within the XPS community. In particular, there is an extensive review by Ernesto Paparazzo entitled “Use and mis-use of x-ray photoemission spectroscopy Ce 3d spectra of Ce₂O₃ and CeO₂” that summarizes the peak fitting methods of different research groups and discusses the challenges associated with the accuracy of peak fitting.

For this study, the analysis of Ce 3d XPS spectra was conducted by our co-author, Mark H. Engelhard, who is an expert in the field of XPS. In our previous work (*Nature* **2022**, 611, 284), we presented a typical XPS peak fit to the Ce 3d spectrum, as shown in **Fig. R1a** below. Similar XPS peak fittings can also be seen in other studies, such as *Nat Commun* **2019**, 10, 1358 and *Nat Catal* **2021**, 4, 469. In this work, the analysis method of Ce 3d XPS spectra follows the same approach as described in the above studies, as shown in **Fig. R1b** (**Fig. 4g** in this work).

We acknowledge that there are other methods used for peak fitting of Ce 3d spectra, and there may be ongoing debates about which method is optimal. However, we believe that the trend of the Ce³⁺/Ce⁴⁺ ratio obtained by our method is consistent with other proposed methods. We appreciate

any additional insights or suggestions the reviewer may have on the proper assignment and deconvolution process. However, we believe that this discussion is beyond the scope of this work.

Figure R1 Typical XPS peak fit of the Ce 3d spectra from the previous study (a) *Nature* **2022**, 611, 284 (<https://doi.org/10.1038/s41586-022-05251-6>) and (b) from Fig. 4g in this work.

2) I did not see a significant difference in frequency between the IR signals (2094 vs. 2089 cm^{-1} , Fig. 1i) of CO adsorbed on two different types of Pt single atoms in Pt/CeO₂ and Pt_{at}CeO₂.

Reply: Although the difference in the IR frequencies of CO adsorbed on the two Pt₁ measured at 100 °C might seem insignificant in **Fig. 1i**, it does exist. The shift in the CO vibrational peak is more noticeable at 35 °C, i.e., 2095 vs. 2088 cm^{-1} (**Fig. 3a and 3b**). This phenomenon was also observed in a recent study in *Nat. Commun.* (<https://doi.org/10.1038/s41467-022-34797-2>), where the IR vibrational frequency of Pt₁/CeO₂ calcined at 550 °C is different from Pt_{AT}CeO₂ calcined at 800 °C in the CO-DRIFTS results. Thus, we believe that the IR difference between the two samples can be used to identify the structural differences of Pt₁ between Pt₁/CeO₂ and Pt_{AT}CeO₂.

3) The dynamic structural evolution of Pt from charged single atoms to reduced clusters was tracked by CO vibrations that vary in frequency depending on both oxidation states and coordination numbers. The formation of Pt clusters was characterized by low-lying CO bands around 2070 and 2050 cm^{-1} (Fig. S19). However, in the previous work (Ref. 22) they assigned these CO bands to reduced Pt_1 . This is a key point and should be discussed in more detail based on the recent study of structure changes of Pt/CeO₂, where IR vibrational bands were observed for CO molecules adsorbed on surface sites exposed by both Pt and ceria support (modified via interaction with Pt, see *J. Phys. Chem. C* 2022, 126, 9051).

Reply: While CO-DRIFTS is powerful tool for distinguishing single atoms from nanoparticles, there is still some controversy surrounding its use. In CeO₂-supported Pt catalysts, the CO bands located around 2090 and 2100 cm^{-1} are generally considered to be the CO adsorbed on single-atom Pt_1 , while the shoulders around 2070 and 2050 cm^{-1} are believed to be due to the formation of Pt clusters (*Angew. Chem. Int. Ed.* **2022**, 61, e202112640; *Science*, **2015**, 350, 189; *ACS Catal.* **2019**, 9, 3978–3990; *Angew. Chem.* **2020**, 132, 20872; *J. Am. Chem. Soc.* **2020**, 142, 169; *Nat Commun* **2020**, 11, 4240; *Nat Commun* **2019**, 10, 3808; *Nat Commun* **2019**, 10, 1358; *ChemCatChem* **2020**, 12, 1726, etc.). However, the assignment of CO bands around 2070 and 2050 cm^{-1} should also be validated by other techniques, such as TEM, XPS, as there is evidence to suggest that similar shoulder may also be present for CO adsorption on low-valence Pt_1 (*Nat. Mater.* **2019**, 18, 746).

In our previous work (Ref. 22, *Angew. Chem.* **2021**, 60, 26054), Pt₁/CeO₂ synthesized by thermal shock method at an extremely high temperature of 1500 K did show CO bands around 2070 and 2050 cm^{-1} . However, other characterization results such as TEM and XPS (Fig. S12, S15 in Ref. 22) ruled out the existence of Pt clusters. In contrast, the TEM images (**Fig. 4a,4d**), XPS (**Fig. S23**), and Raman spectra (**Fig. S24**) in this work clearly demonstrate the presence of Pt clusters. Thus, it is crucial to combine CO-DRIFTS data with other characterization techniques to make definitive assignments. As suggested by the reviewer, we have included discussions after **Fig. S19**

to explain why the CO bands around 2070 and 2050 cm^{-1} in this work are attributed to the formation of Pt clusters.

It should be noted that CO adsorption on ceria support is very weak, and the corresponding infrared vibrational bands can only be observed at low temperatures. For example, in the paper mentioned by the reviewer (J. Phys. Chem. C 2022, 126, 9051), the IR spectra were collected at only 113 K (-160 °C). Above room temperature, the observed CO bands can only be assigned to various Pt sites, as seen in the references mentioned in the previous paragraph.

4) The IR band at 2114 cm^{-1} was assigned to CO bound to oxidized PtO_x clusters. However, a closer analysis shows that the 2114 cm^{-1} signal is proportional in intensity to the one at about 2170 cm^{-1} (Fig. 3a-c), revealing that they are more likely to originate from the CO gas phase contribution, rather than CO- PtO_x .

Reply: We appreciate the reviewer for bringing this to our attention. Upon a thorough examination of the temperature-dependent DRIFTS spectra presented in **Fig. 3a-c**, we have determined that the infrared band at 2114 cm^{-1} observed at 120 °C is likely due to the contribution of gas-phase CO, as suggested by the reviewer. Therefore, we have removed the label from **Fig. 3a** and relevant discussion from the manuscript. However, we want to emphasize that this does not affect the primary conclusion of our work, as there is still a distinct IR shoulder for Pt clusters that can be observed and analyzed.

5) The redispersion of Pt NPs into Pt_1 in O_2 at 500 °C was visualized by HAADF-STEM. It would be better to mark them in the corresponding images (Fig. S27).

Reply: The redispersed Pt_1 atoms after oxidation treatment at 500 °C are now labeled in high-resolution HAADF-STEM in **Fig. 4b,4e**. Thank you for the suggestion!

6) I would suggest to perform additional calculations on the CO vibrational frequency. The agreement of computed and experimental frequencies of CO adsorbed on different Pt₁ sites would strengthen the conclusions, in particular the proposed structure of adsorbed Pt₁ on ceria.

Reply: We have included additional calculations of the CO vibrational frequency on different Pt₁ structures. During the CO oxidation, the Pt-O coordination number in both catalysts decreases. Therefore, we compared the CO vibrational frequency on adsorbed and square-planar PtO_x (x = 1 ~ 4) with different Pt-O coordination numbers, as shown in **Fig. R2**. We found that CO adsorbed on the square-planar Pt₁O₃ structure has a CO vibrational frequency of 2090.5 (step site) and 2078.4 cm⁻¹ (surface), which is similar to the experimental results in **Fig. 3b** (~2088 cm⁻¹). In contrast, CO adsorbed on the adsorbed Pt₁O₃ structure has a CO vibrational frequency of 2100.6 cm⁻¹, which agrees with the results in **Fig. 3a** (2101 cm⁻¹ at 35°C and 2095 cm⁻¹ at 80°C). By comparing the CO vibrational frequencies obtained from CO-DRIFTS and theoretical calculation, we can conclude that the observed CO-Pt₁ band in CO-DRIFTS should be related to the adsorbed CO on PtO₃. We have added supplementary calculations in Fig. S33 in the supporting information and relevant discussion in the section of “Theoretical insight into the dynamic behaviors”. With these theoretical calculation results, the adsorbed Pt₁ structure in Pt/CeO₂ can be further confirmed. Thank you for this valuable suggestion.

Figure R2 Vibrational frequency of adsorbed CO on various PtO_x structures.

7) It has been reported that differently shaped ceria NPs could undergo massive surface restructuring under activation and reaction conditions at elevated temperatures. The morphological changes of ceria support should have significant impact on the interaction with Pt and the activity for CO oxidation (see e.g. J. Phys. Chem. Lett. 2020, 11, 7925).

Reply: We agree with the reviewer that surface reconstruction of CeO₂ at different calcination temperatures (500 °C and 800 °C) may cause morphological changes that further affect the interaction between Pt and the supports. To minimize these effects, the CeO₂ support was pre-calcined at 800 °C for 10 h. A small amount of Pt (0.1 wt%) was then impregnated on the synthesized 800CeO₂ support to maintain single-dispersed Pt₁ nature, followed by calcination at 500 and 800 °C to obtain 0.1Pt/800CeO₂ and 0.1Pt_{AT}800CeO₂, respectively. Since the support was pre-calcined at 800 °C, the two samples exhibited similar porosity properties (**Fig. R3a**) and CeO₂

particle size (Fig. R3b) as the 800CeO₂ support, indicating that the 800CeO₂ support is stable after loading Pt at different calcination temperatures. We assume that there are no significant structural changes of CeO₂ in 0.1Pt/800CeO₂ and 0.1Pt_{AT}800CeO₂ compared to the 800CeO₂ support. We found that the activity of 0.1Pt/800CeO₂ was still significantly higher than that of 0.1Pt_{AT}800CeO₂ (Fig. S14), similar to Pt/CeO₂ and Pt_{AT}CeO₂ (Fig. 2a). Therefore, we believe that the initial Pt_I local structure is mainly responsible for the CO oxidation activities in this work. The relevant data are presented in Fig. S14-S18, and the discussion has been incorporated after Fig. S14.

Figure R3 (a) N₂ adsorption-desorption isotherms at 77 K and (b) XRD patterns of 800CeO₂, 0.1Pt/800CeO₂ and 0.1Pt_{AT}800CeO₂. The average particle size of CeO₂ in (b) was determined by Scherrer equation from CeO₂ (111) peak.

Reviewer #2: This paper reports the dynamic properties of Pt/CeO₂ single atom catalysts according to their initial calcination step (500°C for Pt/CeO₂ and 800°C for Pt_{AT}CeO₂). The Dynamic evolution of the Pt nanostructure was probed under CO oxidation condition and redox sequences by advanced in-situ characterization techniques (EXAFS, DRIFT, RAMAN) and DFT calculations. Noteworthy results deal with the initial Pt nanostructure (Pt single atoms interaction with ceria surface) which was found to depend on the initial annealing temperature. Pt nanoparticles formed by a reduction step redisperse under oxidizing conditions and return to their initial nanostructure.

(1) However, these main findings are not original as dynamic properties of Pt/CeO₂ are well documented in the literature including the initial nanostructure of Pt/CeO₂ single atoms as a function of the annealing temperature. The location of Pt single atoms on ceria after a calcination at low (400-500°C) and high (750-800°C) has been already fully described by similar characterization techniques and DFT (see for instance : 10.1002/anie.201402342, <https://doi.org/10.1038/s41929-020-00508-7>, <https://doi.org/10.1021/acs.jpcc.2c02420>, <https://doi.org/10.1016/j.apcatb.2020.119304>).

Reply: Although CeO₂ supported Pt₁ catalysts have been extensively studied using various characterization techniques, some important questions remain unanswered, particularly under reaction conditions. Firstly, the debate on whether Pt₁ on CeO₂ will transform into Pt clusters under CO oxidation conditions is still ongoing. Some researchers argue that Pt₁ is rigid (<https://doi.org/10.1038/ncomms10801>), while others believe that Pt₁ is mobile and can sinter into Pt clusters under CO oxidation conditions (<https://doi.org/10.1038/s41929-020-00508-7>). The question is whether the two types of Pt₁ in Pt/CeO₂ and Pt_{AT}CeO₂ change in the same or different ways under reaction conditions. Secondly, no studies have been reported on why the two Pt₁ catalysts, calcined at different temperatures, exhibit drastically different activities.

In this work, we combined *in situ* characterization techniques and DFT calculations to address these questions. Our findings show that Pt/CeO₂, calcined at 500 °C, has an adsorbed Pt₁ structure

that is mobile and transforms into Pt clusters above 100 °C under CO oxidation conditions. In contrast, Pt_{AT}CeO₂ calcined at 800 °C has a square planar Pt₁ structure with a stronger Pt-O interaction and cannot sinter under CO oxidation conditions at the elevated temperatures. Therefore, the real active sites in Pt/CeO₂ and Pt_{AT}CeO₂ are the *in-situ* formed Pt clusters and partially reduced Pt₁, respectively. During the reduction of both Pt/CeO₂ and Pt_{AT}CeO₂ in CO, Pt₁ transforms to Pt nanoparticles. However, after undergoing the same re-oxidation treatment at 500 °C, the two reduced samples reverted to different Pt₁ states, similar to their respective initial structures. These main findings are original and have not been reported in previous studies.

(2) The reversibility between single atoms and nanoparticles as a function of temperature and redox atmosphere and its consequence on CO oxidation catalytic activity is also well reported in previous papers (see for instance : 10.1002/anie.200803126, 10.1039/d0cy00732c).

Reply: Previous papers have reported on the reversibility between single atoms and nanoparticles in response to changes in temperature and redox atmosphere, as well as their related CO oxidation activity. However, our work focuses specifically on how the initial state of the Pt single atoms, i.e., adsorbed (Pt/CeO₂) and square-planar Pt₁ (Pt_{AT}CeO₂) synthesized at 500 and 800 °C, respectively, affects the reactivity and reversibility - a topic that has not been previously studied.

To address these questions, we reduced Pt/CeO₂ and Pt_{AT}CeO₂ in CO at 275 °C, and found that both reduced samples showed similar CO oxidation activity (**Fig. 2a**) due to the formation of Pt clusters. We then recalcined the two reduced samples in O₂ at 500 °C to study their dispersion behavior. While the redispersion behavior is known, the redispersed Pt₁ structure is unclear. Our finding revealed that the reduced Pt_{AT}CeO₂ redispersed into square-planar Pt₁ after recalcination in O₂ at only 500 °C, while the reduced Pt/CeO₂ redispersed to adsorbed Pt₁. These two different Pt₁ structures exhibited distinct reactivity in CO oxidation, as we explained in the response to comment #1. We believe that the observed memory behavior of Pt₁ is unique.

(3) One of the main difficulties with Pt/CeO₂ catalysts is to differentiate oxidized Pt clusters and Pt single atoms using STEM or EXAFS. The presence of few oxidized Pt clusters can strongly modify the surface reactivity of the catalyst. From my point of view, it is difficult to clearly conclude from the results of the characterization techniques used in this study that the investigated catalysts only contain adsorbed single atoms and no oxidized Pt clusters. Furthermore, the Pt-Pt coordination numbers determined by EXAFS are not reported. In a recent study (10.1021/jacs.9b09156), Resasco et al. have demonstrated that isolated Pt atoms cannot chemisorb CO at low temperature, contrary to the catalysts of this study (Fig 1i). The same paper and another recent study (<https://doi.org/10.1021/acs.jpcc.1c06784>) have assigned bond CO with a stretch at 2095 cm⁻¹ to adsorption on oxidized Pt clusters. Similar IR bands were observed in this study at 2094 cm⁻¹ for Pt/CeO₂ and to a lesser extent at 2089 cm⁻¹ for Pt_{AT}CeO₂ at 100°C under CO oxidation condition but were attributed to ionic Pt although they could indicate the presence of small oxidized Pt clusters, especially on the catalyst calcined at 500°C.

Reply: We agree that if only a few oxidized Pt clusters are present in Pt/CeO₂, they may evade detection in STEM or EXAFS. However, we can rule out the influence of oxidized Pt clusters on fresh Pt/CeO₂ for the following reasons:

(1) Resasco et al. (10.1021/jacs.9b09156) claim in their CO-DRIFTS study that “Pt_{iso} species with significant coordination to oxygen from the support should bind CO weakly”. This conclusion is based on a computational study (*Phys. Chem. Chem. Phys.* 2016, 18, 22108), which considers some fully coordinated Pt₁ atoms on CeO₂ surface. We fully agree with these computational results. Theoretically, it is not surprising that CO is hardly adsorbed on Pt₁ that is already fully bonded with oxygen. However, fully coordinated Pt₁ (which is stable in air) will transform into less-coordinated Pt₁ under CO oxidation conditions, as evidenced by the decrease in the first-shell Pt-O coordination number (Fig. 3f,3g) after switching the exposing atmosphere from air to CO + O₂. Our DFT calculation in Fig. 5 also demonstrates similar behavior. Therefore, we cannot simply say that single-atom Pt₁ does not adsorb CO. Furthermore, the temperature-dependent CO-DRIFTS in Fig.

3a,3b show that the CO band intensity initially increases with the increase of temperature. If the IR band is ascribed to adsorbed CO on Pt clusters, a higher peak intensity should be obtained at the lower temperature of 35 °C. Instead, the above phenomenon can be explained by the fact that the formation of less-coordinated Pt₁ under CO oxidation condition is favored at high temperatures, which is able to adsorb CO. According to the reviewer's reference (<https://doi.org/10.1021/acs.jpcc.1c06784>), the authors did not attribute the bonded CO stretch at 2095 cm⁻¹ to adsorption on oxidized Pt clusters. The original statement reads "it is possible that the former was related to Pt²⁺ SAC (as proposed by us) or actually oxidized PtO_x clusters". The authors did not exclude the possibility of Pt₁, and multiple studies have associated this band around 2090 cm⁻¹ to CO adsorption on Pt₁ (*Science* **2016**, 353, 150; *Science* **2017**, 358, 1419; *ACS Catal.* **2019**, 9, 3978; *ACS Catal.* **2019**, 9, 8738; *Nat Commun* **2019**, 10, 1358; *Nat. Catal.* **2020**, 3, 824; *Energy Environ. Sci.* **2020**, 13, 4903; *Angew. Chem. Int. Ed.* **2020**, 59, 20691; *Angew. Chem. Int. Ed.* **2021**, 60, 5240; *Angew. Chem. Int. Ed.* **2021**, 60, 4038; *ACS Catal.* **2021**, 11, 8701; *Angew. Chem. Int. Ed.* **2022**, 61, e202112640; *Nature* **2022**, 611, 284; *J. Am. Chem. Soc.* **2022**, 144, 21255; *Nat Commun* **2022**, 13, 7070, etc.). Therefore, it is not justified to attribute the 2090 cm⁻¹ peak to Pt clusters. Additionally, the weak peak intensity between 2~3 Å in EXAFS (**Fig. 3f,3g**) and the proximity of characteristic peaks of Pt-Ce make it difficult to fit the Pt-Pt coordination numbers precisely under *in situ* EXAFS conditions in CO and O₂.

(2) To address the reviewer's concerns, we synthesized two catalysts using the same preparation methods with only 0.1 wt% Pt loading: 0.1Pt/CeO₂ and 0.1Pt_{AT}CeO₂. We present the CO oxidation activity and characterizations in **Fig. S14-S18**. The obtained 0.1Pt/CeO₂ is still significantly more active than 0.1Pt_{AT}CeO₂. Further discussion on this can be found after **Fig. S14**. Considering the low Pt loading, it is unlikely for oxidized Pt clusters to exist in fresh 0.1Pt/CeO₂ after calcination at 500 °C in air.

(3) Pt clusters are detected only in Pt/CeO₂ above 100 °C during CO oxidation conditions (**Fig. 3a,3h**). These Pt clusters, which are formed *in situ*, are considered as the active site for CO

oxidation. It is likely that in fresh Pt/CeO₂, there are only very few Pt clusters present below the detection limit, which should be significantly less than the Pt clusters formed *in situ*. Thus, even if very few Pt clusters exist, that are not expected to affect the activity of Pt/CeO₂.

(4) In addition, Fig. S13c clearly shows a small band near 2072 cm⁻¹, characteristic of Pt clusters as observed after CO reduction (Fig S19). Another insight of the possible presence of PtO_x clusters in Pt/CeO₂ is its activation energy for CO oxidation which was found to be 44.6 kJ/mole. This value corresponds to reduced catalysts containing Pt nanoparticles (see for instance 10.1021/jacs.9b09156, 10.1039/d0cy00732c). The presence of small PtO_x clusters in Pt/CeO₂ could also explain its good catalytic activity (Fig. 2a).

Reply: You are correct, and we need to distinguish between two states: (1) the initial Pt structure in fresh samples, and (2) the actual Pt structure during the reaction. We have concluded that the initial Pt structure is single-dispersed in both Pt/CeO₂ and Pt_{AT}CeO₂. However, Pt₁ in Pt/CeO₂ can be partially reduced under CO oxidation condition (CO: O₂ = 1: 4, lean condition) below 100 °C, and eventually sintered to Pt clusters after further heating above 100 °C. The *in situ* formation of Pt clusters can be evidenced by *in situ* characterizations such as the shoulder at 2070 cm⁻¹ in CO-DRIFTS (**Fig. 3a**) and a new cluster signature in XPS (**Fig. 3h**). From the light-off curve of Pt/CeO₂ (**Fig. 2a**), the formation temperature of Pt clusters is similar to the onset temperature of CO oxidation. Therefore, we conclude that the *in situ* formed Pt cluster under reaction conditions is responsible for the enhanced CO oxidation activity in Pt/CeO₂. In contrast, Pt₁ in Pt_{AT}CeO₂ cannot sinter under CO oxidation conditions at elevated temperatures, although Pt₁ can be partially reduced according to *in situ* XANES (**Fig. 3e**), EXAFS (**Fig. 3g**), XPS (**Fig. 3i**), and other analyses. The low apparent activation energy of 44.6 kJ/mol of Pt/CeO₂ is similar to that of Pt clusters mentioned by the reviewer, further demonstrating the *in situ* formation of Pt clusters under CO oxidation conditions. In **Fig. S13c**, the small band around 2072 cm⁻¹ at 100 °C in Pt/CeO₂ is attributed to CO adsorbed on Pt clusters. The initial CO/O₂ ratio used in this CO-DRIFTS was 1,

unlike the 1:4 ratio used in **Fig. 1i** and **Fig. 2**. We used various CO/O₂ ratios to study the effect of O₂ partial pressures on CO adsorption. Clearly, the CO-rich reaction condition (CO: O₂ = 1) would induce the formation of Pt clusters in Pt/CeO₂ at the lower reaction temperature of 100 °C in **Fig. S13c**. This is reasonable because Pt₁ is stable in O₂ but not in CO, and more CO is expected to favor the formation of Pt clusters. Overall, Pt₁ in Pt/CeO₂ can form Pt clusters under CO oxidation conditions at elevated reaction temperatures.

(5) According to the authors, the Pt single atoms in Pt/CeO₂ would be easier to sinter and to form Pt nanoparticles in reducing atmosphere than ones in Pt_{AT}CeO₂. If so, why the two catalysts achieve the same activity for CO oxidation after the same reduction step in CO? By the same way, Fig. S26 seems to indicate that the redispersion of Pt is slower on Pt_{AT}/CeO₂. Is it coherent with the proposed mechanism?

Reply: It should be noted that Pt₁ in Pt/CeO₂ is more prone to sintering under CO oxidation conditions, rather than reducing conditions. In a reducing atmosphere (275 °C for 20 mins), both Pt/CeO₂ and Pt_{AT}CeO₂ catalysts showed complete reduction of Pt₁ to Pt nanoparticles, as observed in TEM (**Fig. 4a,4d**), CO-DRIFTS (**Fig. S19**), and XPS (**Fig. S23**), resulting in similar activity (**Fig. 2a**) for both reduced catalysts. This indicates that Pt₁ cannot survive under reducing conditions in these two samples.

Under CO oxidation conditions, Pt₁ in Pt/CeO₂ is more easily reduced, as shown in CO-DRIFTS (**Fig. 3a,3b**) and XANES (**Fig. 3d,3e**). In **Fig. S26**, it can be concluded that Pt nanoparticles in reduced Pt/CeO₂ and Pt_{AT}CeO₂ can redisperse even in O₂-rich CO oxidation conditions (CO: O₂ = 1:4) up to 500 °C. The redispersion rate is slower in reduced Pt_{AT}CeO₂, possibly due to kinetic limitations in the migration of Pt atoms to specific locations to form a square-planar structure. However, both reduced samples can easily redisperse upon direct treatment in O₂ at 500 °C (**Fig. 4c,4f**). Therefore, there is no conflict between this redispersion behavior and our proposed mechanism.

(6) I would recommend before considering this paper for publication to clearly confirm the absence of any PtO_x clusters on Pt/CeO_2 by CO pulse chemisorption and H_2 -TPR. This latter technique could give an insight on the presence of PtO_x clusters from the onset reduction temperature as isolated Pt^{2+} sites are inactive towards H_2 dissociation (10.1021/acscatal.8b00330, 10.1039/c6cp00627b, <https://doi.org/10.1021/acscatal.1c04565>). According to J. Resasco et al. (10.1021/jacs.9b09156), catalysts in which Pt is reduced at low temperatures (around 200 °C) contain PtO_x clusters.

Reply: We thank the reviewer for the constructive and thought-provoking questions, and we appreciate the opportunity to address them. We acknowledge the reviewer's concern that characterization techniques may not detect all Pt clusters in fresh samples. However, under CO oxidation conditions ($\text{CO}:\text{O}_2 = 1:4$), Pt clusters are formed *in situ* from Pt_1 in Pt/CeO_2 above 100 °C (**Fig. 3a,3h**). The *in situ* formed Pt clusters can be detected and considered as the active site in Pt/CeO_2 for CO oxidation. Even if there are very few Pt clusters present in fresh Pt/CeO_2 below the detection limit, the content of these clusters should be much less than the Pt clusters formed *in situ*. Furthermore, experiments with low Pt loading (0.1 wt%) had similar experimental results (**Fig. S14-S18**). At such a low Pt loading, we believe that there are no Pt clusters in fresh sample after calcination in air at 500 °C. Even if there are, they do not affect the activity too much because of the *in situ* formed Pt clusters.

Regarding the reviewer's suggestions, we have conducted both CO pulse chemisorption and H_2 -TPR experiments. For CO pulse chemisorption (**Fig. R1**), about 24.4% and 2.3% of Pt_1 in Pt/CeO_2 and $\text{Pt}_{\text{AT}}\text{CeO}_2$, respectively, can adsorb CO on average. Since the initial fully-coordinated Pt_1 cannot adsorb CO until it is reduced to the less-coordinated Pt_1 , the more CO adsorption on Pt/CeO_2 at room temperature, the easier it is for Pt_1 to be reduced. This is in agreement with our DFT calculation in **Fig. S32**, where O vacancy of adsorbed PtO_x is much easier to form. However, this technique cannot completely rule out the existence of few Pt clusters. Based on the H_2 -TPR (**Fig. R2**), Pt/CeO_2 and $\text{Pt}_{\text{AT}}\text{CeO}_2$ have similar reduction temperature around 200 °C.

In a recent study by *Nat. Commun.* (<https://doi.org/10.1038/s41467-022-34797-2>), it was also observed that Pt₁/CeO₂-550 calcined at 550 °C shows much higher CO oxidation activity than Pt₁/CeO₂-800 calcined at 800 °C. This study found that Pt is single-dispersed in both samples. However, the sintering of Pt₁ into Pt clusters under reaction conditions was not considered in Pt₁/CeO₂-550, which is one of the core aspects of our work.

Overall, we believe that fresh Pt/CeO₂ after synthesis has no Pt clusters, and the in situ formed Pt clusters are the active sites for CO oxidation in Pt/CeO₂.

Figure R1 CO pulse chemisorption for Pt/CeO₂ and Pt_{AT}CeO₂ at 25 °C.

Figure R2 Temperature-programmed reduction of H₂ of CeO₂, 800CeO₂, Pt/CeO₂ and Pt_{AT}CeO₂.

Reviewer #3: This manuscript investigates the reversible structural evolution of Pt/CeO₂ and Pt_{AT}CeO₂ SACs under different atmospheres. In addition, the two SACs catalysts display dramatically different catalytic activity towards CO oxidation, which are explained by the different initial Pt₁ local configurations. However, considering some catalytic behavior of the SACs, the identification of the active site and the related structural-activity relationship is suspicious, which is needed to be addressed in details.

Reply: We appreciate the thoughtful and constructive comments from the reviewer, which have helped to improve the quality of the manuscript. In order to provide a more comprehensive discussion on the identification of the active site and the structural-activity relationship, we have addressed the reviewer's comments in the following responses

(1) P13, Line 310: The processing temperatures for Pt/CeO₂-CO and Pt/CeO₂-CO-O₂ are not shown in the Methods section.

Reply: The reduction and oxidation temperatures are 275 and 500 °C respectively, which has been added in the Methods section. Thanks for pointing this out.

(2) For Pt_{AT}CeO₂ sample treated at 800 °C, the particle size of CeO₂ increases from 12 to ~22 nm with the significantly decreased surface area. In addition to the surface reconstruction, the agglomeration of bulk phase CeO₂ is inevitable. If it is possible that a large amount of Pt enters the lattice of CeO₂? This will affect the amount of available Pt atoms on the surface.

Reply: This is a good question. Based on the XRD patterns in **Fig. S2**, there is no noticeable shift of diffraction peaks of CeO₂ in Pt_{AT}CeO₂ compared to pure CeO₂ and Pt/CeO₂. In addition, the surface Pt contents of Pt/CeO₂ and Pt_{AT}CeO₂, as determined by XPS, are 0.8% and 0.9% (**Table S1**). If a large amount of Pt entered the CeO₂ lattice in Pt_{AT}CeO₂, an obvious shift of XRD diffraction peaks and a decrease in surface Pt content would have been expected. Furthermore, if Pt/CeO₂ and Pt_{AT}CeO₂ were treated in CO at 275 °C, the two reduced catalysts exhibited similar

reactivity (**Fig. 2a**). If a large amount of Pt were inside the lattice of CeO_2 , the surface Pt clusters obtained after reduction of $\text{Pt}_{\text{AT}}\text{CeO}_2$ should be less than those in reduced Pt/CeO_2 , which would cause lower activity. Therefore, the low reactivity of $\text{Pt}_{\text{AT}}\text{CeO}_2$ is not considered due to less exposed Pt atoms on the surface.

Figure R1 (a) N_2 adsorption-desorption isotherms at 77 K and (b) XRD patterns of 800CeO_2 , $0.1\text{Pt}/800\text{CeO}_2$ and $0.1\text{Pt}_{\text{AT}}800\text{CeO}_2$. The average particle size of CeO_2 in (b) was determined by Scherrer equation from CeO_2 (111) peak.

To further prove this, CeO_2 support was pre-calcined at 800 °C for 10 h. A small amount of Pt (0.1 wt%) was then impregnated on the above-synthesized 800CeO_2 support to maintain single-dispersed properties, followed by calcination at 500 and 800 °C to obtain $0.1\text{Pt}/800\text{CeO}_2$ and $0.1\text{Pt}_{\text{AT}}800\text{CeO}_2$, respectively. Because of the pre-calcination of the support at 800 °C, $0.1\text{Pt}/800\text{CeO}_2$ and $0.1\text{Pt}_{\text{AT}}800\text{CeO}_2$ exhibited similar porosity properties (**Fig. R1a**) and CeO_2 particle size (**Fig. R1b**) as the 800CeO_2 support. These indicate that there was no sintering of 800CeO_2 particles after calcining Pt-impregnated samples at both 500 and 800 °C. It was found

that the activity of 0.1Pt/800CeO₂ was still significantly higher than that of 0.1Pt_{AT}800CeO₂ (**Fig. S14**), similar to Pt/CeO₂ and Pt_{AT}CeO₂ (**Fig. 2a**). Therefore, we can rule out the effect of surface Pt density on the reactivity, since Pt atoms should all stay on the surface in this case. The relevant data are presented in **Fig. S14-S18**, and the discussion has been incorporated after **Fig. S14**.

(3) In Fig. 1i, considering the spectral resolution (4 cm⁻¹), the different IR band is too small (5 cm⁻¹). In addition, why the CO-DRIFT spectra were collected at 100 °C under reaction conditions? The CO atmosphere and room temperature should be better for the characterization.

Reply: The difference in the IR frequencies of CO adsorbed on the two Pt₁ measured at 100 °C is indeed insignificant in **Fig. 1i**. However, it does exist. The shift in the CO vibrational peak becomes more pronounced at 35 °C, i.e. 2095 vs. 2088 cm⁻¹ (**Fig. 3a,3b**). A similar phenomenon was also observed in a recent work in *Nat. Commun.* (<https://doi.org/10.1038/s41467-022-34797-2>), where the IR vibrational frequency of Pt₁/CeO₂ calcined at 550 °C was lower than that of Pt_{AT}CeO₂ calcined at 800 °C in the CO-DRIFTS results.

DRIFTS were collected under CO oxidation rather than pure CO for the following reasons. The reaction atmosphere is a mixed gas of CO and O₂, and it is preferable to study the Pt structure changes under reaction conditions. Additionally, it is well known that the Pt structure changes when the CO+O₂ atmosphere is switched to pure CO. For CO adsorbed on stable structures, CO adsorption at lower temperature is favored. However, the initial Pt₁ structure in Pt/CeO₂ and Pt_{AT}CeO₂ cannot chemisorb CO due to saturated Pt-O coordination, as explained in response to question 3 from reviewer 2. Partial removal of the surrounding oxygens of Pt₁ (**Fig. 3d,3e**) at elevated temperatures promotes CO adsorption. As a result, CO-DRIFTS were collected at different temperatures (35, 80, 100, 120, 150, 180 °C) in this work, as shown in the **Fig. 3a,3b,1i**. However, the peak intensity of adsorbed CO is weaker at 35 °C for both samples. To compare with a relatively stronger peak intensity, CO-DRIFTS at 100 °C were conducted as shown in **Fig. 1i**.

(4) According to the evaluation results, the Pt NPs show significantly better catalytic activity than SACs. Therefore, confirming the real active site in the two fresh SACs is very important for the following structure-activity relationship. As shown in Fig. 2a and Fig. S14, Pt/CeO₂ and 0.1Pt/CeO₂ show significant different light-off temperatures (100 vs. 200 °C). At 200 °C, the CO conversion is 100% over Pt/CeO₂. If the active sites are the same for these two SACs catalysts, the CO conversion should be larger than 10% for 0.1Pt/CeO₂, which is accurately ~2%. This indicates that the active site in Pt/CeO₂ is more active. How does the author comment on this result?

Reply: Thank you for the nice question! The Pt₁ structure in both Pt/CeO₂ and Pt_{AT}CeO₂ changes with temperature under CO oxidation condition, as shown in the *in situ* CO-DRIFTS (**Fig. 3a,3b**), XANES (**Fig. 3d,3e**), EXAFS (**Fig. 3f,3g**), XPS (**Fig. 3h,3i**) etc. Briefly, Pt₁ in Pt/CeO₂ is partially reduced under CO oxidation condition below 100 °C but still maintains the single-dispersed property (**Fig. 3a,3d,3f**). However, Pt₁ transforms to Pt clusters once the temperature is higher than 100 °C (**Fig. 3a, 3f**). From the light-off curve of Pt/CeO₂ (**Fig. 2a**), the formation temperature of Pt clusters is similar to the onset temperature of Pt/CeO₂. Therefore, we conclude that the *in situ* formed Pt clusters under reaction condition are responsible for the activity in Pt/CeO₂.

In contrast, Pt₁ in Pt_{AT}CeO₂ cannot transform to Pt clusters under CO oxidation conditions at the elevated temperatures, although Pt₁ will be partially reduced according to *in situ* XANES (**Fig. 3e**), EXAFS (**Fig. 3g**), XPS (**Fig. 3i**) etc. That is why Pt/CeO₂ is much more active than Pt_{AT}CeO₂.

Regarding the reviewer's comment, Pt/CeO₂ and 0.1Pt/CeO₂ have the same initial Pt₁ structure, which is the adsorbed Pt₁. However, the initial Pt₁ structure is not the real active site under reaction condition. Pt₁ will sinter into Pt clusters above 100 °C, and the *in situ* formed Pt clusters are the real active sites. Since surface Pt density of 0.1Pt/CeO₂ is lower than that of Pt/CeO₂, mobile Pt₁ is expected to be less likely to sinter in 0.1Pt/CeO₂. Therefore, we found a lower activity per Pt atom in 0.1Pt/CeO₂.

In response to this nice comment, we included a sentence after **Fig. S14**: “It should be noted that CO conversion is almost 100% over Pt/CeO₂ at 200 °C (**Fig. 2a**), but only around 2% over 0.1Pt/CeO₂. If the initial Pt₁ structure in Pt/CeO₂ was the real active site, we should expect around 10% conversion over 0.1Pt/CeO₂ at 200 °C. The much lower reactivity in 0.1Pt/CeO₂ might further prove that the *in situ* formed Pt clusters are the real active sites in Pt/CeO₂, since the lower surface Pt₁ density in 0.1Pt/CeO₂ is less likely to sinter under the same condition.”

(5) Similar results are shown over Pt_{AT}CeO₂ and 0.1Pt_{AT}CeO₂ (Figure 2a and S14), the light-off temperatures are 200 and 300 °C, respectively. In addition, the catalytic activity of 0.1Pt_{AT}CeO₂ is similar with that over CeO₂ (Figure S10). Therefore, the Pt cluster and CeO₂ are important active sites for the fresh Pt/CeO₂ and Pt_{AT}CeO₂ catalysts, which are not well considered in this manuscript.

Reply: We have combined the light-off curves for 800CeO₂, Pt_{AT}CeO₂ and 0.1Pt_{AT}CeO₂ in the **Fig. R2a** below. We agree that CeO₂ support contributes to the partial reactivity for Pt_{AT}CeO₂ and 0.1Pt_{AT}CeO₂ (**Fig. R2a**). However, the catalytic activities of Pt_{AT}CeO₂ and 0.1Pt_{AT}CeO₂ are still higher than that of pure CeO₂ (**Fig. R2a**). Even with only 0.1 wt% Pt on the CeO₂ surface, the difference in activity is evident, suggesting that the square-planar Pt₁ in Pt_{AT}CeO₂ is indeed more active than CeO₂. Nevertheless, CeO₂ does contribute to partial activity above 300 °C.

To exclude the influence of CeO₂, we obtained the light-off curves of Pt_{AT}CeO₂ and 0.1Pt_{AT}CeO₂ by subtracting the activity of CeO₂, as shown in **Fig. R2b**. The CO conversion over Pt_{AT}CeO₂ is roughly 10 times higher than that of 0.1Pt_{AT}CeO₂, for example, 48.4% and 4.6% at 347 °C, respectively (**Fig. R2b**). In Pt_{AT}CeO₂, the influence of CeO₂ is not significant, especially below 300 °C (**Fig. R2b**), which does not affect the activity comparison of Pt/CeO₂ and Pt_{AT}CeO₂.

In situ characterization results indicate that square-planar Pt₁ in Pt_{AT}CeO₂ cannot transform into Pt clusters under reaction condition. In contrast, the adsorbed Pt₁ in Pt/CeO₂ partially transforms

into Pt clusters above 100 °C under CO oxidation conditions (Fig. 3a,3d,3h), contributing to near 100% CO conversion at ~200 °C (Fig. 2a). Therefore, the CeO₂ support has a negligible effect on the activity of Pt/CeO₂. The order of activity is as follows: Pt clusters (*in situ* formed from adsorbed Pt₁ in Pt/CeO₂) > square-planar Pt₁ in Pt_{AT}CeO₂ > CeO₂.

We have added the following figures as the new Fig. S15 and included discussions about the effect of CeO₂ support after Fig. S15.

Figure R2 (a) CO oxidation performance of 800CeO₂ (precalcined at 800 °C), 0.1Pt_{AT}CeO₂, and Pt_{AT}CeO₂; (b) CO oxidation performance of 0.1Pt_{AT}CeO₂, and Pt_{AT}CeO₂ after subtracting the activity of CeO₂.

REVIEWERS' COMMENTS

Reviewer #1 (Remarks to the Author):

The authors have adequately responded to most of my concerns. I have only few minor comments below.

1) and 2), The answers are sufficient.

3) The strong interaction between Pt and ceria support could also leads to a substantial modification in structural and electronic properties of ceria surfaces, which has been demonstrated by the IR band splitting of the CO species weakly bound to different surface Ce sites using a low-temperature IR approach. This surface-sensitive and non-destructive approach enables to characterize the intrinsic properties by identifying all surface and interfacial sites exposed by both Pt and ceria support. A short discussion is required regarding the Pt-CeO₂ interaction and the Pt-induced Ce-CO band splitting as reported in the literature.

4) to 6), The answers are sufficient.

7) As regards the thermally driven restructuring of ceria surfaces, I would suggest to include a short paragraph in the main text to mention the impact of ceria surface morphologies.

Reviewer #3 (Remarks to the Author):

The authors have improved the article and I support its publication in the current form.
Please increase the resolution of Figure 1d.

Reviewer #1: The authors have adequately responded to most of my concerns. I have only few minor comments below.

1) and 2), The answers are sufficient.

Reply: Thanks!

3) The strong interaction between Pt and ceria support could also leads to a substantial modification in structural and electronic properties of ceria surfaces, which has been demonstrated by the IR band splitting of the CO species weakly bound to different surface Ce sites using a low-temperature IR approach. This surface-sensitive and non-destructive approach enables to characterize the intrinsic properties by identifying all surface and interfacial sites exposed by both Pt and ceria support. A short discussion is required regarding the Pt-CeO₂ interaction and the Pt-induced Ce-CO band splitting as reported in the literature.

Reply: To address the reviewer's comment, we have included one sentence in the discussion of DRIFTS part and cited the mentioned literature as: *Presence of Pt₁ on CeO₂ surface could induce the splitting of CO-Ce peak, which only exists below room temperature.*³¹

4) to 6), The answers are sufficient.

Reply: Thanks!

7) As regards the thermally driven restructuring of ceria surfaces, I would suggest to include a short paragraph in the main text to mention the impact of ceria surface morphologies.

Reply: To address the reviewer's comment, we have added a couple of sentences regarding the restructuring of CeO₂ surface in the main text and cited the relevant reference as: *It has been reported that surface reconstruction of CeO₂ at different calcination temperatures would affect the catalytic activity.*²⁵ *To minimize these effects, the CeO₂ support was pre-calcined at 800 °C for 10 h to yield 800CeO₂, followed by deposition of 0.1 wt% Pt (to maintain the atomically dispersed*

nature) at 500 and 800 °C to yield 0.1Pt/800CeO₂ and 0.1Pt_{AT}800CeO₂, respectively. Since the support was pre-calcined at 800 °C, these two samples exhibited similar porosity properties (**Fig. S17, Table S3**) and CeO₂ particle size (**Fig. S18**) as the 800CeO₂ support. We found that the activity of 0.1Pt/800CeO₂ was still significantly higher than that of 0.1Pt_{AT}800CeO₂ (**Fig. S14**), similar to Pt/CeO₂ and Pt_{AT}CeO₂ (**Fig. 3a**).

Reviewer #3: The authors have improved the article and I support its publication in the current form. Please increase the resolution of Figure 1d.

Reply: The resolution of Figure 1d has been improved. Thanks!